# Convergence in phosphorus constraints to photosynthesis in forests around the world

**David S. Ellsworth** [1] ✉, **Kristine Y. Crous** [1], **Martin G. De Kauwe** [2,3], **Lore T. Verryckt**[4], **Daniel Goll** [5,6], **Sönke Zaehle**[7], **Keith J. Bloomfield**[8], **Philippe Ciais** [5], **Lucas A. Cernusak** [9], **Tomas F. Domingues**[10], **Mirindi Eric Dusenge**[11,12], **Sabrina Garcia** [13], **Rossella Guerrieri** [14], **F. Yoko Ishida**[9], **Ivan A. Janssens** [4], **Tanaka Kenzo** [15], **Tomoaki Ichie**[16], **Belinda E. Medlyn** [1], **Patrick Meir**[17,18], **Richard J. Norby** [19], **Peter B. Reich** [1,20,21], **Lucy Rowland**[12], **Louis S. Santiago** [22], **Yan Sun**[5,23], **Johan Uddling**[11], **Anthony P. Walker**[24], **K. W. Lasantha K. Weerasinghe**[25], **Martine J. van de Weg**[18], **Yun-Bing Zhang** [26], **Jiao-Lin Zhang** [26] & **Ian J. Wright**[1,27]

Tropical forests take up more carbon (C) from the atmosphere per annum by photosynthesis than any other type of vegetation. Phosphorus (P) limitations to C uptake are paramount for tropical and subtropical forests around the globe. Yet the generality of photosynthesis-P relationships underlying these limitations are in question, and hence are not represented well in terrestrial biosphere models. Here we demonstrate the dependence of photosynthesis and underlying processes on both leaf N and P concentrations. The regulation of photosynthetic capacity by P was similar across four continents. Implementing P constraints in the ORCHIDEE-CNP model, gross photosynthesis was reduced by 36% across the tropics and subtropics relative to traditional N constraints and unlimiting leaf P. Our results provide a quantitative relationship for the P dependence for photosynthesis for the front-end of global terrestrial C models that is consistent with canopy leaf measurements.

Tropical forests contain the majority of the world's higher plant species and absorb a gross of over 35 Pg carbon (C) per annum from the atmosphere, more than any other biome[1]. Forests are key modulators of global climate by virtue of their large C exchange with the atmosphere[2,3]. Amongst global forests, tropical forests comprise the most productive biome per unit land area on Earth in spite of occurring on highly weathered, nutrient-poor soils[1–4]. Notwithstanding the key role played by the tropics in the global carbon cycle[5] and hence for offsetting fossil fuel C emissions, there have been persistent uncertainties in predictions of primary productivity for these forests[1,6,7]. The uncertainties in gross primary productivity arise from a paucity of relevant data on, and understanding of, tropical forest gross photosynthesis and C cycles and their regulation by nutrients, and particularly phosphorus (P)[8–10]. Leaf photosynthetic capacity is the primary driver of C-uptake, and so it is crucial to accurately represent it in terrestrial biosphere models to enable C-cycle predictions for the present-day, and with global change[11,12].

Leaf nitrogen (N) has long been assumed the single, critical nutrient driving variation in photosynthetic capacity[13] and is widely incorporated as a modulator of gross and net photosynthesis in terrestrial biosphere models[14,15]. Most leaf N is invested in ribulose-1,5-biphosphate carboxylase/oxygenase (Rubisco; EC 4.1.1.39), the primary carboxylating enzyme in the Calvin-Benson cycle and central to present-day net photosynthesis. At about one-quarter of terrestrial canopy N, Rubisco is the single most abundant enzyme on Earth[16]. A further quarter of leaf N is allocated to the thylakoid membrane-bound proteins crucial for net photosynthesis[17], resulting in a strong N-dependence of photosynthesis amongst plants around the world.

The strong functional relationship between carboxylation capacity ($V_{cmax}$, in $\mu mol\ CO_2\ m^{-2}$ leaf $s^{-1}$) and leaf N[11,13] (Supplementary Fig. 1) is harnessed as a key driver in most large-scale gross photosynthesis models[18]. However, emerging evidence over the past decade has suggested that the strong relationship of net photosynthesis with leaf N is diminished when leaf phosphorus (P) concentrations are low[11,19]. This important effect is not yet considered in most terrestrial biosphere models (TBMs) including those central to the terrestrial C cycle[20,21].

Alongside N, P has been identified as a second critical element to plant function worldwide[22,23] but its role in photosynthetic capacity is debated. Unravelling the constraint to net photosynthesis by leaf P has been difficult[24,25]. This difficulty is linked to the broad range of specific biological roles played by P in plants, in various compounds such as adenosine triphosphate (ATP), nucleotides and nucleic acids, sugar phosphates, and phospholipids to regulate and support photosynthesis. Theory[26] and biochemical analyses[24,27] have suggested that P deficiency reduces the electron transport capacity of leaves ($J$, in $\mu mol$ electrons $m^{-2}$ leaf $s^{-1}$; Supplementary Fig. 1) and reduces the supply of P to regenerate the key substrate for carboxylation and the Calvin-Benson cycle, ribulose-diphosphate (RuBP). As a result of the roles of N and P associated with different biochemical components controlling photosynthesis (Supplementary Fig. 1), a potential imbalance between the capacity for carboxylation versus electron transport supporting RuBP regeneration could arise in species with low P status or with high leaf N:P ratios, for example on low P soils. Alternatively, if plants maintain a similar functional balance between components of the photosynthetic apparatus, irrespective of soil nutrient concentrations, this would lead to the same ratio of maximum electron transport and RuBP regeneration ($J_{max}$) to carboxylation ($V_{cmax}$) across a range of N and P concentrations in leaves[28]. This functional balance would explain why the Rubisco enzyme carboxylation capacity for photosynthesis is also sometimes associated with P[11,29] even though there is no explicit role of P in carboxylation[26]. With a paucity of data involving chronically low P sites, few studies have fully examined this functional balance hypothesis for photosynthetic components[28,29] across different soils.

Nearly every major large-scale TBM of the C cycle incorporates some form of the Farquhar-von Caemmerer-Berry (FvCB) photosynthesis model[30], which implicitly assumes that N is the primary nutrient limiting photosynthesis[12]––an assumption we here term the "single nutrient-single enzyme" hypothesis (Supplementary Fig. 1). In contrast, few TBMs and only a single model (CABLE-CASA CNP) from the sixth phase of the Coupled Model Intercomparison Project (CMIP6)[21] incorporate a direct role of P in modelled gross photosynthesis[8,9]. Uncertainty about how to represent a general photosynthetic role for P in TBMs stems from considerable variation in reported relationships between photosynthetic biochemistry (e.g. carboxylation capacity, $V_{cmax}$) and leaf P (Supplementary Fig. 2), which may reflect the geographically restricted nature of previous studies, the particular geology of different continental regions[31], and the relatively narrow range of leaf P considered[32].

At the leaf scale there is growing evidence of P-limitation reducing photosynthetic capacity[32–34]. However, broad evidence for a robust P constraint on photosynthesis and its biochemistry with chronic low P availability has been lacking for several reasons. First, regional studies have differed greatly in soil P status and soil orders owing to differences in surface geology and the extent of exposed ancient landscapes differing greatly among continents[35,36]. Thus, they have different degrees of P influence over photosynthesis. Second, the amount of P allocated to key metabolically active compounds that regulate photosynthesis can be highly variable[37]. Coupled with plant species variation in internal allocation of leaf P[37], these uncertainties have impeded inclusion of a robust and general photosynthesis-P relationship into large-scale models[8,9]. They also suggest there could be large variation amongst different studies and continents in the relationship of photosynthetic biochemistry to leaf P.

To address the nature of the P constraint on photosynthetic biochemistry, we analysed how photosynthetic biochemistry, and specifically $V_{cmax}$ and $J_{max}$, varied with leaf P at an unprecedented scale. We compiled a new dataset representing 402 species sampled from 52 sites spanning four continents across the Neotropics, Paleotropics and subtropics (Supplementary Fig. 3 and Supplementary Table 1). These are regions where the role of soil P in regulating productivity is expected to be especially important[22,38]. The dataset includes species from over one-fifth of all known angiosperm families (Supplementary Table 2), encompassing a substantial part of global taxonomic richness and representative of the high plant diversity in the tropics and subtropics (Supplementary Fig. 4). Leaf P concentrations covered a wide range (50-fold) in contrast to previous analyses that spanned about 10-fold in the upper part of the range (Supplementary Fig. 2). We compiled raw data for net photosynthesis responses to $CO_2$ concentration ("$A_{net} - C_i$ curves"), calculating light-saturated photosynthetic rates and biochemical parameters $V_{cmax}$ and $J_{max}$ (electron-transport enabling RuBP regeneration capacity) by inverting the FvCB model[39]. The final dataset consisted of species-at-site mean values ($n = 446$ across sites) for light-saturated maximum net photosynthetic rate ($A_{net}$), $V_{cmax}$, $J_{max}$, and their mass-based quantities ($A_{net\_mass}$, $V_{cmax\_mass}$, $J_{max\_mass}$) as well as leaf N and P concentrations ($N_{mass}$ and $P_{mass}$, respectively), along with a key leaf structural trait, leaf dry mass per area ($M_a$).

Here we demonstrate a dependence of photosynthetic biochemistry ($V_{cmax}$ and $J_{max}$) on leaf N[12,17] but also leaf P on an unprecedented scale, across continents with different underlying soils and parent material geology and diverse plant taxa. We tested for effects of leaf P on $J_{max}$ in accordance with theory[24,25], and employed the new relationships in TBM scenarios for gross primary productivity across the tropics and subtropics to establish how these new relationships affect land-atmosphere $CO_2$ exchange relevant to atmospheric $CO_2$ drawdown across the tropics. Our overall finding is that inclusion of P at the front-end of TBMs for C cycle processes has a large influence on the magnitude of gross $CO_2$ uptake by photosynthesis which supports the incorporation of a robust photosynthesis-P relationships into large-scale terrestrial biosphere models underpinning our assessments of the C cycle[40,41].

## Results

Low leaf P status clearly diminished $V_{cmax}$–N and $J_{max}$–N relationships on a mass basis (Fig. 1a, b and Table 1). We determined this in a regression framework where we defined "low-P" status of plants based on a threshold for leaf P ($P_{mass}$ of $0.92\ mg\ g^{-1}$). Similar relationships held at a range of leaf $P_{mass}$ threshold values (see slopes analysed in Supplementary Fig. 5). The $V_{cmax\_mass}$–$N_{mass}$ slope was twice as steep for "moderate-P" species in this dataset as for low-P species (Fig. 1a; Table 1): for a 5-fold increase in leaf $N_{mass}$, $V_{cmax\_mass}$ of moderate-P species increased 3.3-fold whereas that of low-P species increased just 1.8-fold. Similarly, $J_{max\_mass}$– $N_{mass}$ relationships were nearly 2-fold steeper for moderate-P species than for low-P species (Fig. 1b; Table 1), as were $A_{net\_mass}$–$N_{mass}$ relationships (Table 1).

The $V_{cmax}$–N and $J_{max}$–N data for low P and moderate P species converged at low $N_{mass}$ but at higher $N_{mass}$ the fitted slopes diverged ($P < 0.01$, Table 1). At a leaf $N_{mass}$ of $20\ mg\ g^{-1}$, near the median $N_{mass}$ of our data, both $V_{cmax\_mass}$ and $J_{max\_mass}$ increased by ~40% for species from low to moderate leaf P concentrations (Fig. 1a, b). This clear inhibitory effect of low leaf P on photosynthetic–N relationships was observed when slopes were fit as either least-squares regressions or as standardised major axes (SMA) (Table 1 and Supplementary Table 3; see Methods and Supplementary text).

Relationships between photosynthetic variables and leaf $P_{mass}$ are much less common in the literature than N-based relationships[11,25], but in this dataset P-based relationships were no less significant than the more common relationships with N (Fig. 1c, d). In fact, in this

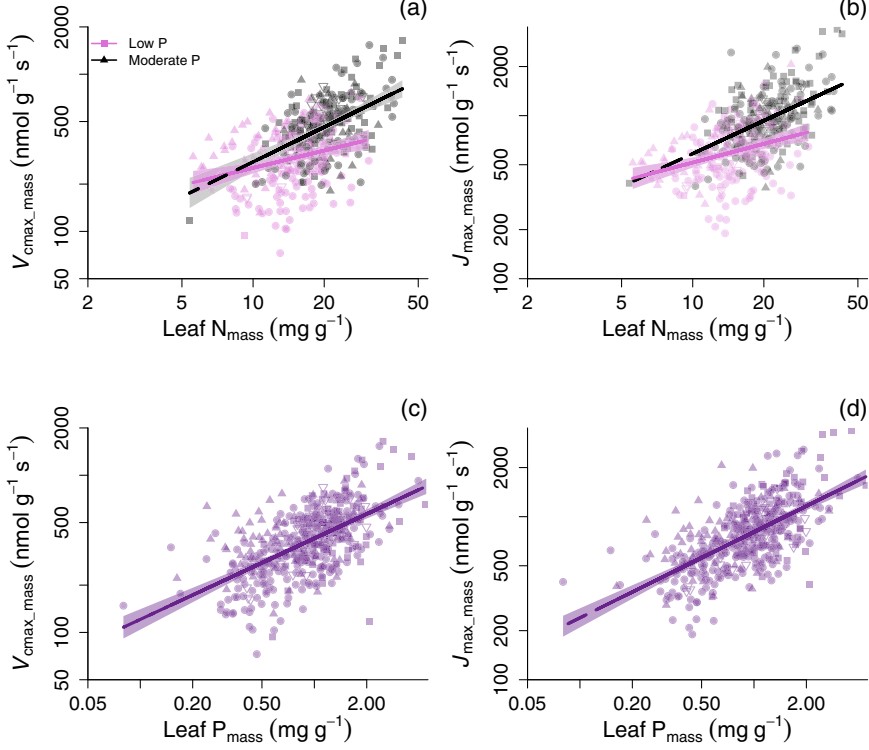

**Fig. 1 | Relationships between leaf photosynthetic characteristics and leaf N and P for diverse woody species across continents. a, b** Relationships between mass-based photosynthetic parameters and leaf N concentration ($N_{mass}$) for tropical and subtropical trees across four continents, for species grouped into two leaf P ($P_{mass}$) classes, "low P" ($P < 0.92$ mg g$^{-1}$) and "moderate P" ($P \geq 0.92$ mg g$^{-1}$). Low P data and lines in **a, b** are plum-coloured, with moderate P species shown as grey and black. Lines are least-squares fits and the shaded areas are the 95% CI regions. Each point represents the mean of a species-site combination, where different symbols of the same colour denote different continents and there are $n = 445$ species-site combinations. **c, d** The relationships between mass-based photosynthetic parameters and leaf phosphorus concentration for tropical and subtropical trees across four continents, with the shaded zone denoting the 95% CI. Least-squares fits and statistics for the lines in **a–d** are shown in Table 1. Photosynthetic parameters are **a, b** leaf mass-based carboxylation capacity normalised to 25 °C ($V_{cmax\_mass}$), and **c, d** leaf mass-based RuBP regeneration capacity normalised to 25 °C ($J_{max\_mass}$).

## Table 1 | Summary of single-factor photosynthetic-nutrient relationships for N and P

| Mass-based | | P status | d.f. | r² | Slope | Intercept | *F*-value | *P* value | *P* value for slope diff. |
|---|---|---|---|---|---|---|---|---|---|
| Depen-dent variable | Ind. variable | | | | | | | | |
| $A_{net}$ | $N$ | Mod. P | 231 | 0.16 | 0.779 | 2.323 | 66.7 | <0.0001 | 0.0013 |
| | | Low P | 212 | 0.19 | 0.369 | 3.242 | 16.3 | <0.0001 | |
| | | All P | 445 | 0.28 | 0.741 | 2.350 | 170.7 | <0.0001 | – |
| $V_{cmax}$ | $N$ | Mod. P | 231 | 0.26 | 0.736 | 3.929 | 79.2 | <0.0001 | 0.0013 |
| | | Low P | 212 | 0.08 | 0.367 | 4.689 | 15.6 | <0.0001 | |
| | | All P | 445 | 0.30 | 0.751 | 3.783 | 194.4 | <0.0001 | – |
| $J_{max}$ | $N$ | Mod. P | 231 | 0.23 | 0.671 | 4.825 | 67.4 | <0.0001 | 0.0075 |
| | | Low P | 212 | 0.10 | 0.382 | 5.366 | 22.0 | <0.0001 | |
| | | All P | 444 | 0.44 | 0.310 | 8.564 | 208.6 | <0.0001 | – |
| $V_{cmax}$ | $P$ | All P | 445 | 0.34 | 0.515 | 5.983 | 231.4 | <0.0001 | – |
| $J_{max}$ | $P$ | All P | 445 | 0.40 | 0.527 | 6.692 | 300.0 | <0.0001 | – |

Slope diff. is respective to P status (low P versus moderate P concentration; see text), d.f. indicates the denominator degrees of freedom. The equivalent area-based results are shown in the standardized major axis analysis in Supplementary Table 3.
Analyses were done using ordinary least-square (OLS) regressions for different P status levels and all P levels together ('All P'). Both the dependent and independent ('Ind.') variables for the least-squares regressions are natural logarithm-transformed. The difference between the low and moderate P status groups are defined in the text according to a $P_{mass}$ threshold of 0.92 mg g$^{-1}$, and the differences in slopes ('slope diff.') were tested using separate-slopes analyses.

predominately pan-tropical dataset (Table 1), leaf $P_{mass}$ on its own generally explained more variation in maximum photosynthesis rates and biochemistry per unit mass ($V_{cmax\_mass}$ and especially $J_{max\_mass}$, Fig. 1c, d) than did leaf N on its own. This demonstrates a strong modulation of photosynthetic biochemical capacity by leaf P for

diverse broadleaved plants. For either mass- and area-based $J_{max}$, the explanatory power (r²) was about 9–13% higher for leaf P than for leaf N, with associated reductions in mean-square errors (Table 2) demonstrating leaf P effects on photosynthesis and the capacity for RuBP regeneration ($J_{max\_mass}$; Fig. 1).

**Table 2 | Summary statistics for multiple regression analyses**

| Dependent variable | Independent variables | d.f. | Inter-cept | Slopes for main effects | Slope for N × P inter-action | Over-all $r^2$ | Whole-model *P* value | Inter-action term *P* value | $M_a$ term *P* value |
|---|---|---|---|---|---|---|---|---|---|
| Mass-based | | | | | | | | | |
| $A_{mass}$ | $N_{mass}$, $P_{mass}$ | 444 | 3.129 | 0.479, 0.271 | – | 0.33 | <0.0001 | – | – |
| $A_{mass}$ | $N_{mass}$, $P_{mass}$ and $N_{mass} \times P_{mass}$ | 443 | 2.998 | 0.513, –0.547 | 0.294 | 0.35 | <0.0001 | 0.0003 | – |
| $A_{mass}$ | $N_{mass}$, $P_{mass}$, $M_a$ and $N_{mass} \times P_{mass}$ | 442 | 7.055 | 0.139, –0.259, –0.626 | 0.161 | 0.45 | <0.0001 | 0.0330 | 0.0001 |
| $M_a$ | $N_{mass}$, $P_{mass}$ and $N_{mass} \times P_{mass}$ | 443 | 6.484 | –0.598, 0.461 | –0.214 | 0.51 | <0.0001 | <0.0001 | – |
| $V_{cmax\_mass}$ | $N_{mass}$, $P_{mass}$ | 444 | 4.780 | 0.415, 0.347 | – | 0.40 | <0.0001 | – | – |
| $V_{cmax\_mass}$ | $N_{mass}$, $P_{mass}$ and $N_{mass} \times P_{mass}$* | 443 | 4.636 | 0.453, –0.546 | 0.321 | 0.42 | <0.0001 | 0.0001 | – |
| $V_{cmax\_mass}$ | $N_{mass}$, $P_{mass}$, $M_a$ and $N_{mass} \times P_{mass}$ | 442 | 7.136 | 0.222, –0.368, –0.385 | 0.239 | 0.46 | <0.0001 | 0.0009 | 0.0001 |
| $J_{max\_mass}$ | $N_{mass}$, $P_{mass}$ | 444 | 5.667 | 0.354, 0.383 | – | 0.45 | <0.0001 | – | – |
| $J_{max\_mass}$ | $N_{mass}$, $P_{mass}$ and $N_{mass} \times P_{mass}$* | 443 | 5.535 | 0.388, –0.436 | 0.295 | 0.47 | <0.0001 | – | – |
| $J_{max\_mass}$ | $N_{mass}$, $P_{mass}$, $M_a$ and $N_{mass} \times P_{mass}$ | 442 | 8.401 | 0.124, –0.232, –0.442 | 0.200 | 0.53 | <0.0001 | <0.0001 | 0.0001 |

d.f. indicates the denominator degrees of freedom.
Regressions showing photosynthesis and mass-based biochemical parameters ($A_{mass}$, $V_{cmax\_mass}$, $J_{max\_mass}$) and leaf mass per area ($M_a$) versus leaf $N_{mass}$ and $P_{mass}$, including their interaction ($N_{mass} \times P_{mass}$). Slopes for main effects are ordered according to the list of independent variables. The $N_{mass} \times P_{mass}$ interactions were positive in all cases except for $M_a$. All tests for interaction and additive terms were done using *F*-tests. All variables are natural-logarithm transformed, and the models for $V_{cmax\_mass}$ and $J_{max\_mass}$ with $N_{mass}$ and $P_{mass}$ are illustrated in Supplementary Fig. 8. The recommended model for TBMs is indicated by *.

We further tested whether $J_{max\_mass}$–$P_{mass}$ slopes fitted to individual continents differed from slopes fitted to the remainder of the dataset. Differences among continents might occur, for example, as soil orders and the predominance of ancient eroded bedrock can differ substantially among regions[23,38]. However, continent-specific slope differences in $J_{max\_mass}$-$P_{mass}$ were not observed (Fig. 2, *P* > 0.1), nor was the $J_{max\_mass}$-$P_{mass}$ slope different for any continent compared to that of the remainder of the dataset (*P* > 0.05, using continent as a covariate; Supplementary Table 4). There were similar results for $V_{cmax\_mass}$-$P_{mass}$ (Supplementary Table 4). Thus the observed relationships between leaf P and photosynthetic biochemistry are robust and convergent across continents. Furthermore, climate parameters were not a significant covariate for these relationships (Supplementary Fig. 6). There were important taxonomic differences exhibited among the species sampled across continents (Supplementary Table 2), a characteristic feature of the diverse species richness in tropical forests. Hence the convergence in photosynthetic biochemistry-$P_{mass}$ relationships across continents was particularly surprising, lending support for the robustness of these relationships and their utility in TBMs. While soil P is generally believed to not routinely limit productivity in northern temperate ecosystems[35,42], a limited dataset from temperate zone Northern Hemisphere analysed in the same manner as our large and diverse cross-continent dataset was combined with the relevant temperate data from TRY[43]. Though still a very limited dataset relative to the tropical and subtropical species were analysed in Fig. 1, results in Supplementary Fig. 7 were broadly consistent with the larger dataset of broadleaved evergreen species in Fig. 1.

Considering $N_{mass}$ and $P_{mass}$ effects on photosynthetic traits in a multivariate regression framework (Table 2) led to conclusions consistent with the above analyses that grouped $P_{mass}$ into low and moderate categories (Fig. 1). That is, we observed statistically significant $N_{mass} \times P_{mass}$ interactions with $V_{cmax\_mass}$ and $J_{max\_mass}$ (interactions in all cases, *P* < 0.001; Table 2, Supplementary Fig. 8), indicating flatter trait relationships at lower $P_{mass}$. Similar results of a reduced slope at low $P_{mass}$ were found when variation in $M_a$ was also accounted for in the model (Table 2). Together, leaf $N_{mass}$ and $P_{mass}$ (and their interaction) explained 43% and 48% of variation in $V_{cmax\_mass}$ and $J_{max\_mass}$, respectively (Table 2). Moderate multicollinearity was observed with a

significant correlation between $N_{mass}$ and $P_{mass}$ ($r^2 = 0.39$, *P* < 0.0001) across the dataset. However, this does not affect predictability of $V_{cmax\_mass}$ or $J_{max\_mass}$ from $N_{mass}$ and $P_{mass}$[44]. On an area basis, $V_{cmax}$–$N_{area}$ and $J_{max}$–$N_{area}$ relationships showed significantly lower intercepts at lower leaf P status (all with *P* < 0.02), rather than differences in slopes (Supplementary Table 3). That is, at any given leaf $N_{area}$, lower $P_{area}$ leaves tended to have lower area-based $V_{cmax}$, $J_{max}$ and $A_{net}$.

As further evidence of leaf P effects on the $J_{max}$ component of photosynthesis, as expected by theory[24,27], there was a reduction in the slope of the relationship between $J_{max}$ and $V_{cmax}$ that depended on leaf P status (leaf $P_{mass}$ classes; Fig. 3). $J_{max}$ was lower in proportion to $V_{cmax}$ for leaves with low $P_{mass}$ than for leaves with high $P_{mass}$, as would be expected by a stronger influence of leaf $P_{mass}$ on the capacity for $J_{max}$ than $V_{cmax}$. This supports the earlier evidence (Figs. 1 and 2) that there are strong effects of leaf P on $J_{max}$, stronger than these effects are for $V_{cmax}$ (Table 1).

Terrestrial models of photosynthesis and GPP need to represent key biological processes. To test whether the central relationships of photosynthetic components with nutrients that we found in Fig. 1 and Table 2 (multiple regressions) could produce large enough effects on canopy photosynthesis to merit consideration in TBMs, we conducted a model experiment involving the ORCHIDEE-CNP model[15] for the tropics and subtropics. We applied three-way $V_{cmax\_mass}$–$N_{mass}$–$P_{mass}$ and $J_{max\_mass}$–$N_{mass}$–$P_{mass}$ relationships (Table 2, Supplementary Fig. 9) in ORCHIDEE-CNP for latitudes <35° N and S to simulate forest canopy photosynthesis under scenarios of unlimited leaf P and limited-leaf P content (see Methods). In the equatorial zone, modelled forest GPP was reduced by 36% to just over 1800 g C m$^{-2}$ y$^{-1}$ in the P-limited scenario. However we found that, assuming an unlimited plant P, gross primary productivity (GPP) in equatorial latitudes was simulated to increase by almost a factor of two, approaching 3000 g C m$^{-2}$ y$^{-1}$ across the region[1,45] (Fig. 4a). The total difference in modelled subtropical- and-tropical GPP comparing the unlimited P with the limited P scenario was large, at roughly 70 Pg C (Fig. 4d; between 33 °N and 33 °S), an over-estimate by roughly half of the expected annual total global C uptake in a year. While the upper simulated range is above observational-constrained estimates[43], there were similar proportional reductions in GPP with reduced P compared with unlimited P across

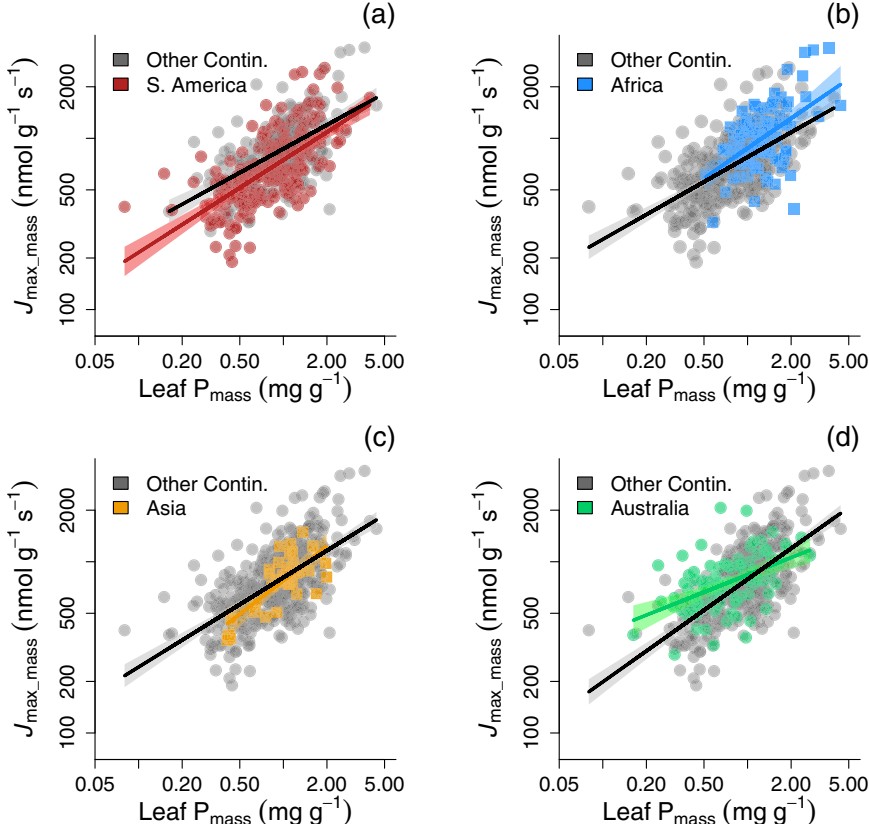

**Fig. 2 | Comparison of $J_{max\_mass}$ as a function of $P_{mass}$ for woody species on different continents.** Each continent is shown compared to the full remainder of the dataset (denoted 'Other Continents'), for **a** South America, (**b**) Africa, (**c**) Asia, (**d**) Australia. Each point denotes a different species-site mean. There was no significant continent effect in the analysis ($P > 0.05$; see Supplementary Table 5). The 95% CI for each relationship around each line is shown in grey for 'Other Continents', and the 95% CI for each continent is shaded in the corresponding colour for each continent being compared.

the tropical domain (e.g., 36% reduction, Fig. 4c). As a result, we confirmed that a front-end control of P over canopy photosynthesis can produce significant alterations in modelled GPP estimates for this TBM.

## Discussion

Considered all together, our results show strong and consistent evidence for a negative effect of low P on photosynthetic biochemistry across a large diversity of woody species, regardless of continent, the basis of expression, or the statistical approach to slope-fitting. The observed effects of both N and P and their interactions on photosynthetic biochemistry (Fig. 1 and Table 2, Supplementary Fig. 8) demonstrate a significant inhibitory effect of low leaf P on photosynthetic biochemistry that is currently captured in few TBMs[8]. These relationships are consistent with recent evidence for P limitation of tree growth in lowland tropical forests[31,46], the declining magnitude of P resorption across the tropics to mid-latitude regions[23] and modelled biomass C uptake and sequestration[15]. Moreover, we clearly show that leaf P affects photosynthetic biochemistry in a way that has not been implemented in previous models[8,15,47,48] but is general across a wide diversity of species, and across continents, supported by extensive field observations.

We demonstrated that such P limitations can arise through reduced biochemical capacity for photosynthesis at low leaf $P_{mass}$ in combination with moderately high leaf $N_{mass}$, likely in concert with low orthophosphate pools for photosynthetic biochemistry[24,25,27]. The similarity in photosynthetic biochemistry-$P_{mass}$ relationships across taxa and continents (Fig. 2 and Supplementary Table 4) occurs in spite of potential differences in how P is allocated to metabolic function across taxa and/or geological substrates on different continents[31,35].

The convergence in these relationships across continents suggests an overall similar and conservative use of P in photosynthesis across a range of soils but largely at low soil P availability. Further work is needed to disentangle changes in the botanical composition of natural vegetation in response to varying soil N and P availability from the response of individual species to contrasting N and P supply.

The proportion of leaf P involved in photosynthesis versus other functions varies among species[37], yet there are still too few data from field-sampled plants to draw solid generalisations about which leaf P fraction is key to regulating photosynthesis. If the fraction of P that is metabolically active varies with total P concentration in leaves, then we would have expected differences in key relationships such as $J_{max\_mass}$–$P_{mass}$ across sites and continents. Instead, the striking convergence in our results from subtropical and tropical sites points to general mechanisms that may be in play for $C_3$ plants from other biomes where low-P soils occur. There is evidence that relationships like $J_{max\_mass}$–$P_{mass}$ are generalizable to Northern Hemisphere temperate woody plants (Refs. 19, 49, and Supplementary Fig. 7) but in this regard there is a clear need for further work involving temperate coniferous and deciduous trees. In fact, in the extensive TRY database, there is a paucity of Northern Hemisphere temperate records (Supplementary Fig. 7), particularly involving species-at-site values for $J_{max}$, $M_a$ and $P_{mass}$ measured together. We identify this as an area for further research, involving both broadleaved and needle-leaved temperate and boreal species.

The cross-continent relationships for $V_{cmax\_mass}$ and $J_{max\_mass}$ with $P_{mass}$ that we have presented in Fig. 1c, d establish an important benchmark in plant physiology, bearing in mind that these relationships are across plant species. Instantaneous photosynthetic P-use efficiency (the ratio of mass-based photosynthesis to leaf P

concentration) has been hypothesized to be high in plant species adapted to survive at very low soil P levels due to a variety of adaptations[50]. Figure 1c, d provide a set of quantitative relationships against which elevated photosynthetic P-use efficiency can be compared to objectively test for enhanced photosynthetic P-use efficiency and enable traits that confer it to be identified.

Future analyses should clarify how components of photosynthetic biochemistry are reduced with chronic, low leaf $P_{mass}$ in contrast to the rapid, acute P deficiency that has previously been examined[24]. With a

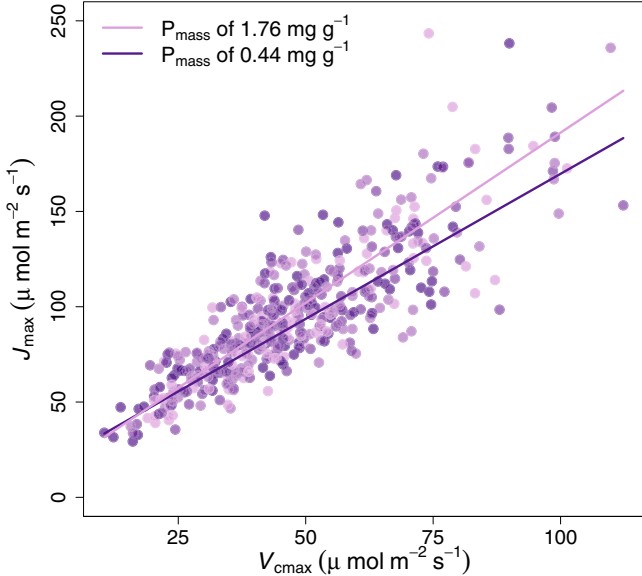

**Fig. 3 | The correlation between $J_{max}$ and $V_{cmax}$ for leaves with different leaf P concentrations.** The colour scheme indicates low leaf $P_{mass}$ in purple shades with increasing $P_{mass}$ corresponding to progressively lighter shades of purple to plum. Each points is a species-site mean. The lines shown are for the two end-member leaf $P_{mass}$ classes: mean low $P_{mass}$ of $0.44 \pm 0.11$ (s.d.), and mean high $P_{mass}$ of $1.76 \pm 0.55$ (s.d.). The OLS regressions shown are: $J_{max} = 17.5 + 1.52 * V_{cmax}$ for low $P_{mass}$ ($r^2 = 0.82$), and $J_{max} = 12.8 + 1.79 * V_{cmax}$ for high $P_{mass}$ ($r^2 = 0.71$), with $P < 0.0001$ for both regressions. $V_{cmax}$ and $J_{max}$ are temperature-normalised to 25 °C (see methods). The slope terms of the lines are significantly different at $P = 0.0355$ using $P_{mass}$ class as a categorical variable in interaction with the independent variable.

paucity of enzyme function work involving tropical species in realistic low-P soil conditions, there can be advances with further physiological and molecular work in tropical regions and species adapted to low-P soils[51] in order to better support the mode of regulation of photosynthetic biochemistry by cellular P supplies.

The mode of action for P suggested by our analyses is likely more complex than the direct, single protein-N paradigm that has existed for Rubisco and other photosynthetic proteins[17], but it is no less important. The larger proportion of variation in the biochemistry of photosynthesis described by $P_{mass}$ versus $N_{mass}$ in Fig. 1, and higher coefficients of determination and lower mean square error for the P-only compared to N-only models (Tables 1 and 2), and the convergence across continents (Fig. 2 and Supplementary Table 4), all indicate a strong functional role for P constraining photosynthesis. Our results demonstrate the existence of consistent, across-continent reductions in the capacity for photosynthesis with low leaf P, cutting across a wide range of higher plant families and involving all vegetated continents (Fig. 2 and Supplementary Fig. 7). This indicates a set of robust relationships that can be incorporated into TBMs for a range of plant functional types.

### Functional balance of the biochemistry of photosynthesis

The $J_{max}/V_{cmax}$ ratio signifies the optimal functional balance between the two fundamental components of photosynthesis: carboxylation versus electron transport and RuBP regeneration. The $J_{max}-V_{cmax}$ connection has been extensively described and analysed[49,52] and is capitalized as a commonly employed short-cut in TBMs[12]. While it is still debated how low P affects photosynthetic biochemistry, some clues emerge from our study. The conventional hypothesis that there is little or no role of P in regulating $V_{cmax}$[11,12] is not supported by evidence here across species and soils (Fig. 1). Moreover, the idea that $V_{cmax}$ is closely coupled to $J_{max}$ and hence $V_{cmax}-P$ relationships are simply a consequence of its control of $J_{max}$ and subsequent functional balance between $J_{max}$ and $V_{cmax}$ is partially but not fully supported by Fig. 3. Instead, acclimation of photosynthesis to low-P environments via adjustments in the $J_{max}/V_{cmax}$ ratio involves a role for P in photosynthetic protein assembly and enzyme activation via phosphorylation[53] which suggests an alternative set of ways that P can influence the state of Rubisco and hence $V_{cmax}$.

A physiological imbalance in the capacity for $J_{max}$ versus $V_{cmax}$ would indicate excess electron transport at light saturation. The rate of

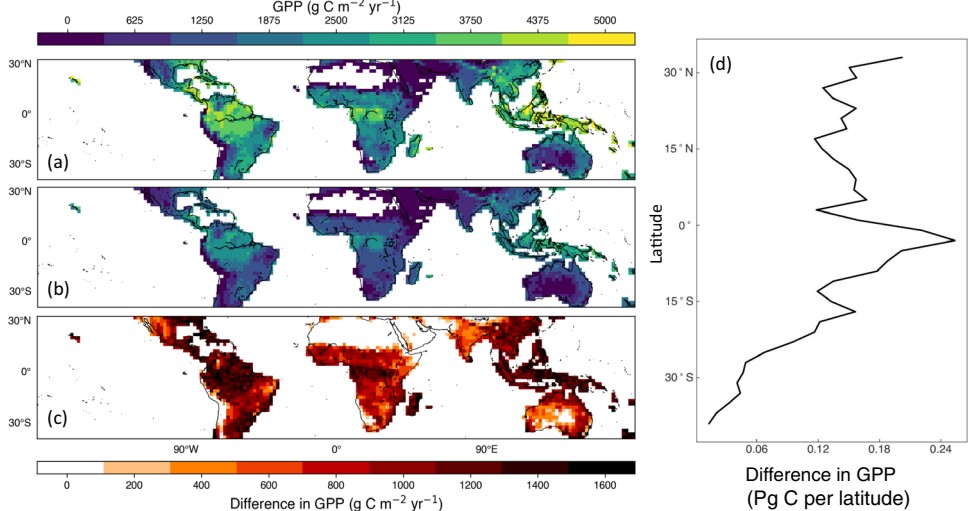

**Fig. 4 | Modelled gross primary productivity (GPP) for tropical and subtropical zones with ORCHIDEE-CNP. a** GPP from ORCHIDEE-CNP simulations assuming N constraints but a high P everywhere (no P constraint). **b** GPP as in **a**, but including P constraints according to a version of the multiple regression in Table 2. The colour scale for **a**, **b** are at top. **c** the difference between GPP from the model with N but not P constraints as shown in **a** and the ORCHIDEE-CNP simulations with P constraints according to **b**, with colour scale at bottom. (**d**) the zonal difference in GPP shown in **c** using 2° latitudinal bands and aggregated across longitudes around the globe.

electron transport $J$ is coupled to proton translocation and generates the trans-thylakoid pH gradient, which drives the regeneration of ATP and RuBP. Excess $J$ would tend to risk excessive stromal acidification which itself would disrupt the electron transport chain and risk damage to the leaf tissue[24]. The opposite situation, with excess carboxylation capacity at suboptimal $J_{max}/V_{cmax}$ ratio, would lead to insufficient ATP supplies to regenerate RuBP[26]. Either phenomenon would tend to shift the capacity toward an optimal balance between the two functional components of the photosynthetic apparatus in leaves at the top of the canopy. Such shifts explain why the $J_{max}/V_{cmax}$ ratio is nearly constant in sunlit leaves across a wide range of situations[28,49]. However, despite the arguments and evidence of constancy for the $J_{max}/V_{cmax}$ ratio[52], the different roles of N and P in primary photosynthetic reactions (Supplementary Fig. 1) implies a small shift in $J_{max}/V_{cmax}$ ratio with respect to lower leaf $P_{mass}$ that is consistent with our observations (Fig. 3). The $J_{max}/V_{cmax}$ shift with increasing $P_{mass}$ in our dataset is not large (e.g., a 10% reduction with low $P_{mass}$; Fig. 3), supporting a functional balance for the components of the photosynthetic apparatus[28]. Still, the majority of TBMs that parameterise $J_{max}$ based on the basis of this functional balance with $V_{cmax}$ and a highly conserved $J_{max}/V_{cmax}$ should consider these changes in the functional balance with low leaf P and high leaf N:P ratio and the mechanistic implications of this in models.

## Model analysis and implications

Given the strong role of tropical $CO_2$ exchange in regulating the earth's C exchange with the atmosphere and hence climate, an influential role for leaf P concentration on photosynthesis would be expected to be manifest at large scales and impact the C cycle. The relationships in Fig. 1 and Table 2 have functional significance and should be considered in TBMs seeking to link nutrient cycles to the C cycle[9,15]. Thus, we utilised the observed mass-based relationships for biochemistry-N and -P in a model analysis and found large proportional reductions in GPP with reduced P compared with unlimited P across the tropical and subtropical domain (Fig. 4c). Our estimate of the reductions in GPP of 36% across this key set of mid- to low-latitude biomes are large. A smaller change in GPP between unlimited leaf P and limited-leaf P scenarios could be possible for a different global model or alternative model implementation of the field results. However, our objective here was to evaluate our new formulation of photosynthetic biochemistry with a leaf $P_{mass}$ dependence and its potential impact on large-scale C cycling, to help refine uncertainties in GPP which are large for the tropical zone[54]. Future efforts should determine the effects this model parameterisation would have on net C storage in different global terrestrial models, recognising that there are a number of downstream processes after gross photosynthesis that could enhance or diminish P effects on net primary productivity at the large scale[6,47,55] compared to GPP as analysed here.

The reduction in GPP emerging from the dependence of $V_{cmax}$ and $J_{max}$ on P results in modelled tropical C uptake estimates is consistent with modelled C cycle outcomes inferred from atmospheric inversion and flux site upscaling models[1,56] (Supplementary Fig. 9). The general and robust relationships of photosynthetic parameters with leaf $P_{mass}$ and $N_{mass}$ (Fig. 2) along with N-P interactions (Fig. 3) could readily be included by other terrestrial biosphere models[8,57,58] either with a P biogeochemistry submodel[40] or with existing global datasets of P spatial variability[59]. This would ensure that GPP was appropriately constrained by both leaf N and P as two major limiting macronutrients around the globe[22,23].

The regulation of photosynthesis by leaf P and its effect on rate-limiting biochemical parameters has demonstrable consequences for large-scale forest C uptake (Fig. 4) with important implications for understanding the C cycle not only for the tropics but for low-P sites around the world. From our findings of consistent P constraints on photosynthetic biochemistry across continents, we argue that there is

no longer any basis for ignoring the P effects on photosynthesis in TBMs even if P is not uniformly low throughout the tropics[23,58]. Also, the P constraints implemented here for photosynthesis in the ORCHIDEE-CNP model (Fig. 4) illustrate the effects on GPP but do not address more complex ecosystem C cycle processes that can be sensitive to soil P such as biomass allocation, growth, forest structure and leaf area[8,9]. For instance, species compositional changes along P availability gradients are an additional way in which forest productivity may be modulated by P[46]. These phenomena have been proposed for modelling[8,9] or are already implemented in TBMs[55], but the photosynthetic biochemistry proposed here is a key front-end control on the tropical and subtropical C cycle.

In TBMs that underlie our predictions of future carbon sink behaviour, projections of C uptake for the tropics and the rate of climate forcing by $CO_2$[8] have remained unconstrained by leaf or soil P status[8,9], likely biasing GPP predictions for these regions. These effects are particularly important given the role of tropical and subtropical regions in regulating global $CO_2$ uptake and vegetation-climate interactions. Given that there are stable relationships for photosynthesis with broad range of leaf N and P across continents, global terrestrial C cycle models can now represent both nutrient constraints on net photosynthesis and its biochemical determinants to improve NPP predictions.

## Methods

### Leaf gas exchange

We compiled 17,913 data points for controlled photosynthetic responses to [$CO_2$] for a set of pan-tropical sites involving published and unpublished raw data (Supplementary Table 1) that were measured using standard techniques[60] and similar instrumentation (Li-6400, Li-Cor Inc.). Mean annual precipitation at these sites varied widely, from 500 to 3000 mm y$^{-1}$, as did mean annual temperatures (10–30 °C, Supplementary Table 1). Data were analysed and fit for biochemical parameters in a common framework[30,39]. The data we assembled represent the most comprehensive analysis of photosynthetic biochemistry across plant families (Supplementary Fig. 4) measured through 2019. The published data sources are Refs. 61–72 (Supplementary Table 1) with raw data for these studies as well as the unpublished data sources in Supplementary Table 1 compiled together (see Data availability statement). Climate data not available from direct measurements at the sites were estimated based on gridded climate data[73]. Only naturally occurring trees, shrubs and lianas at mature life stages were included in the data. Gaining access by construction-style cranes, leaves were sampled at considerable heights (20 m to 70 m above the ground) at six of the sites (Lambir Hills National Park, Sarawak, Malaysia; Bubeng, China; San Lorenzo National Park, Panama; Parque Natural Metropólitano, Panama; EucFACE, NSW, Australia; and Cape Tribulation, Queensland, Australia). For the remainder of the studies ($n$ = 46 sites) leaves were sampled on branches that had been collected from mid-to-upper canopy positions, placed in water and recut to maintain a viable water supply. All leaves were identified by data contributors as "sunlit" to represent photosynthetic function in the sunlit portion of the tree canopy. We required analyses of P concentrations as well as N concentrations for the dataset (Supplementary Fig. 10). Multiple individuals were measured for most of the species, and these data were averaged for a species-at-site average that was used in the analyses ($n$ = 471 species-site values, $n$ = 446 complete with both $N_{mass}$ and $P_{mass}$). Unlike previous such analyses[29,32], we focused our analysis on species-level variation, given that species described the largest source of variation in leaf P[74] (Supplementary Fig. 11). This approach also avoided excessive weight given to particular species that were represented by many multiple sample leaves in the analyses and minimised the possibility of finding statistically significant correlations due to a large number of data points but with low predictive power[61]. A small, limited dataset from Europe and North America that

was compiled from direct measurements and from the TRY database[43] analysed in the same manner as our large and diverse cross-continent dataset (Supplementary Fig. 8) to compare the results with deciduous and gymnosperm species (five species of *Quercus* and *Pinus*, as major Northern Hemisphere genera).

To ensure that drought did not confound our results, measurements were collected as much as possible during the wet season or the early part of the dry season. Also, leaves that showed very low stomatal conductances (<30 mmol $H_2O$ $m^{-2}$ $s^{-1}$) were removed from the analysis as photosynthetic metabolism in such cases could either be limited by low nutrients or low conductance to $CO_2$ diffusion and hence would not be diagnostic for low N and P concentrations. We also ensured that $A_{mass}$ > 20 nmol $g^{-1}$ $s^{-1}$ and the curve-fit CV < 30% for the initial slope as criteria for inclusion to the overall dataset, consistent with previous analyses of photosynthetic capacity[19,60,75]. As leaf $P_{mass}$ and other variables were approximately log-normally distributed, we transformed these variables appropriately in the analyses. In some analyses we treated leaf P as a covariate and grouped species into two leaf P classes: "moderate P" and "low P", based on a threshold corresponding to the median leaf P concentration in the dataset (leaf $P_{mass}$ of 0.92 mg $g^{-1}$) similar to what was used previously[19]. In so doing, we recognize that low leaf $P_{mass}$ may not always reflect soil availability due to species-level mechanisms that can affect P uptake[50], though leaf $P_{mass}$ is most relevant to leaf internal physiology.

We based our analyses on mass-based photosynthetic parameters to enhance predictive capacity to use these relationships in modelling. However, the relationships examined also included area-based quantities such as $V_{cmax}$ and $J_{max}$, and our findings are generally as applicable to area-based measures as mass-based ones (Supplementary Table 3). We note as in many other analyses that have been done that area-based least-squares regression fits are often significant but are weaker than the mass-based ones[76]. Leaf mass per area ($M_a$), the conversion factor between area- and mass-bases of expression, was also included as a covariate in a subset of multiple regression analyses (Table 2). In such cases the fitted coefficients for leaf N or P effects on photosynthetic traits can be thought as being independent from the basis of expression (i.e. area vs mass, Ref. 76; Supplementary Note 1).

## Data fitting and statistical analyses

The curve fitting used the *plantecophys* package[39] for least-squares minimisation in R. The fits were obtained using an inversion of the FvCB biochemical model of leaf photosynthesis[30] which is employed in the land surface portion of the World Climate Research Programme's CMIP6 models[77] and many other land surface models[21]. Enzyme kinetic constants are used to compute $V_{cmax}$ and $J_{max}$ normalised to 25 °C according to functions representing acclimation and adaptation of photosynthetic temperature response kinetics[78] using site temperatures summarised in Supplementary Table 1 and Ref. 73. They are reported in all figures as temperature-normalised to 25 °C. We made common assumptions about the kinetic coefficients for the Rubisco enzyme and biophysical constants in the model for all species apart from these photosynthetic Arrhenius temperature response parameters in Ref. 78. We assumed an infinite mesophyll conductance term in the analysis, so rates are expressed on an apparent basis. Assuming a finite mesophyll conductance equal to the species mean stomatal conductance that was measured did not qualitatively change the results or findings.

Our statistical analyses were conducted using natural logarithm-transformed data as appropriate for nutrients per unit dry mass. Bulk leaf P concentrations were used since few studies, and less than 2% of the data here, have analysed P fractions in leaves. The main set of analyses use ordinary least-squares (OLS) regression fits to species-level data, with examination of residual and quantile (Q-Q) plots to ensure the models met assumptions of the technique. In addition, we used standardised major axis analyses as a secondary supporting

approach that avoids undue bias in slope estimates[79] for Fig. 1 (see Supplementary text). Differences in slopes in OLS regression analyses were tested using multiple regression with the appropriate categorical variable (e.g., continent, N:P ratio class, etc.) as a categorical variable along with the independent variable in an interaction model. The significance of the interaction effect was used to test for separate rather than parallel slopes[44]. In Table 2, we provide appropriate functions for estimating leaf biochemistry parameters $V_{cmax\_mass}$ and $J_{max\_mass}$ depending on $N_{mass}$ and $P_{mass}$ that can be used in TBMs. A dependence of $M_a$ on $P_{mass}$ (Table 2) should be used to convert the estimates to an area basis.

To examine the balance between $J_{max}$ and $V_{cmax}$ and test if there was a dependence on leaf $P_{mass}$, we conducted a multiple regression involving $V_{cmax}$ as independent variable and $P_{mass}$ as covariate. $P_{mass}$ was highly significant in the model ($P = 0.00129$), indicating different $J_{max}$–$V_{cmax}$ relationships with leaf $P_{mass}$. To visualise this, we further divided "low P" and "moderate P" status classes (from Fig. 1) each in half, and then tested for the difference between these classes using the two outermost extremes of $P_{mass}$ ($P_{mass}$ of 0.44 mg $g^{-1}$ and $n = 111$ versus 1.76 mg $g^{-1}$, $n = 112$ observations). The two outer $P_{mass}$ classes showed significantly different ($P = 0.035$) slopes for $J_{max}$–$V_{cmax}$ relationships using $P_{mass}$ class as a categorical variable in interaction with the independent variable.

## Model and analysis of pan-tropical P-limitations

We used the land surface model ORCHIDEE-CNP version 1.1[40,80]. The model simulates the terrestrial biogeochemical cycles of C, N and P and their interactions as well as the water budget and the exchanges of energy, water and $CO_2$ and N between the atmosphere and the biosphere. ORCHIDEE-CNP version 1.1 is well evaluated at site-level, including nutrient dynamics and their effects on tropical gas exchange[55]. The model is able to reproduce (1) the shift from N to P limited plant growth along a soil formation chronosequence in Hawaii[40] and (2) gas exchange measurements on P-poor tropical soils[55].

To understand the role of low P in restricting gross primary productivity based on the leaf-level responses we identified, we replaced the original N dependency of photosynthesis in the model with a new relationship based on Fig. 3. The relationships used were $V_{cmax\_mass}$ = exp(4.4490 + 0.3472*ln($P_{mass}$) + 0.49078*ln($N_{mass}$)) and $J_{max\_mass}$ = exp(5.4944 + 0.3735*ln($P_{mass}$) + 0.4144*ln($N_{mass}$)), which were converted to area-basis using the $M_a$ predicted in the model. Subsequent to the modelling, data was added from two other sites (Manaus, Brazil and Bubeng, China; $n = 28$ species added), which did not appreciably or quantitatively affect the results (see newer relationships, Supplementary Fig. 7). The area-based versions of these functions were $V_{cmax}$ = exp(4.308 + 0.298*ln($P_{area}$) + 0.197*ln($N_{area}$)) and $J_{max}$ = exp(5.139 + 0.325*ln($P_{area}$) + 0.112*ln($N_{area}$)) based on fits to the raw data. The ORCHIDEE-CNP model here widened the range that leaf N:P ratio can vary from the original narrow range that was predicted in the model (N:P from 16.7–22.6 in Ref. 68) to an N:P range of 5–60, corresponding to the 25th and 75th percentiles of the N:P in the leaf photosynthesis dataset. As a result of the wider range of leaf N:P ratios, we replaced the scaling function for plant P acquisition processes (biochemical mineralisation and root uptake, Ref. 40) by a sigmoidal function [$f$($PN_{plant}$)] and chose the coefficient such that processes sharply increases between plant labile N:P ratio of 15 to 25:

$$f\left(PN_{plant}\right) = \frac{1}{\left(1 + \exp\left(\left(-\frac{N:P}{2}\right) + 10\right)\right)} \quad (1)$$

The N:P ratio of 15 roughly corresponds to the ratio where plant communities have shown a switch from N to P limitation[23,81] and we chose this ratio as the middle of the co-limitation range.

With the modified model, we performed pan-tropical and subtropical simulation on a 2° × 2° spatial resolution using the simulation

protocol in Ref. 82. The protocol takes into account historic changes in land cover, $CO_2$ concentration, climate, and N and P deposition since 1860 (called 'experiment S1' in Ref. 82). Climate forcing was derived from the CRUNCEP v.7 meteorological dataset (National Centers for Environmental Prediction-National Center for Atmospheric Research (NCEP-NCAR) and Climatic Research Unit-University of East Anglia). For the simulations, first the cycles of C, N and P were brought into equilibrium (<1% in global stocks) with the boundary conditions of 1860. Second, the simulation was continued to 2012 using time series of land cover (SYNMAP), climate (CRUNCEP7), atmospheric deposition, fertilizer and $CO_2$ concentration (NOAA GLOBALVIEW-$CO_2$ dataset). For the analysis, we used the average GPP over a 21-year period (1992–2012) to represent the present-day productivity of grid cells and evaluate how the mathematical formulation for photosynthesis involving P would affect vegetation GPP.

We considered unlimited P supply to be when leaf N:P ratio was 5, and limited P when leaf N:P ratio was estimated from the model by a P biogeochemistry submodule[40]. This implementation might have overestimated the P limitation effect on GPP, but was done to demarcate P-limited and non-limited photosynthesis. We diagnosed the P-unlimited GPP in the simulations by using photosynthetic parameters, $J_{max}$ and $V_{cmax}$, which correspond to the computed leaf $N_{mass}$ but assuming a maximum leaf $P_{mass}$ calculated from the minimum N:P ratio of 5 g N (g P)$^{-1}$. Based on these photosynthetic parameters we recalculated GPP for the conditions (water, light, leaf area index) at each time step. The estimated GPP did not affect state variables and, thus, the feedback between GPP and LAI is not accounted for in the calculation of potential GPP. Subsequent to the modelling, data was added from two other sites, but the relationships with N and P remained similar to those used in the modelling (Fig. 3, Table 2). The relationships used by the model did not use the N × P interaction term (Table 2). Nonetheless, we found a less-pronounced effect of leaf P on GPP by about 15% with these relationships compared to the relationships based on the complete dataset shown in Table 2, so the model results we report in Fig. 4b–d are slightly more conservative than if we had implemented the relationships from the full dataset.

## Reporting summary

Further information on research design is available in the Nature Research Reporting Summary linked to this article.

## Data availability

The photosynthesis and leaf nutrient data reported in the paper are available at https://doi.org/10.6084/m9.figshare.20010485.v1, and the model results are available on the European open-access repository Zenodo at https://doi.org/10.5281/zenodo.6619615. All other data reported in the paper are presented in the supplementary materials.

## Code availability

The R code used for analyses is at https://github.com/ellswor2/photo_p_repo2.git. The source code for ORCHIDEE is at https://doi.org/10.14768/20200407002.1.

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

## Acknowledgements

This work was supported in part by grants from the Australian Research Council Discovery scheme (grants DP160102452 and DP210100115) and the NSW Research Attraction and Acceleration Program (independent grants to D.S.E. and M.D.K.). D.S.E. also acknowledges research fellowships through the Chinese Academy of Sciences President's International Fellowship Initiative, Grant No. 2018VBA0015, and the German Academic Exchange program (DAAD). M.D.K. acknowledge support from the Australian Research Council Centre of Excellence for Climate Extremes (CE170100023), the ARC Discovery Grant (DP190101823). I.J.W. acknowledges support by Australian Research Council (DP170103410). Y.S., P.C., D.S.G., L.T.V. and I.A.J. are funded by the "IMBALANCE-P" project of the European Research Council (ERC-2013-SyG-610028). A.P.W. was also supported by the US DOE, Office of Science, Office of Biological and Environmental Research at Oak Ridge National Laboratory, which is managed by UT-Battelle, LLC, for the US DOE under contract DE-AC05-00OR22725. J.L.Z. received funding from the National Natural Science Foundation of China (31870385) and the CAS 135 program (2017XTBG-F01). K.J.B., T.F.D., F.Y.I. and P.M. were supported by the UK National Environment Research Council 'Tropical Biomes in Transition (TROBIT)' consortium via research grant NE/D01185x/1 to the University of Edinburgh. T.F.D. and S.G. received funds from USAID for funding via the PEER program (grant agreement AID-OAA-A-11-00012). The contribution of P.R. was supported by the U.S. NSF Biological Integration Institutes grant DBI-2021898.

## Author contributions

D.S.E., K.Y.C., M.D.K. and I.J.W. designed the research, and D.S.E. compiled the raw dataset with the data contributions. D.S.E. coordinated the project with input from K.Y.C., I.J.W., M.D.K. and L.T.V., K.B., L.C., K.Y.C., T.D., M.E.D., S.G., R.G., I.A.J., B.E.M., P.M., R.J.N., L.R., L.S., T.K., T.I., J.U., L.V., A.W., M.v.W., Y.-B.Z and J.-L. Z. contributed data. M.D.K., S.Z., D.S.E., Y.S. and D.G. designed the simulation model runs for Orchidee. The model development analysis was done by D.G and Y.S. D.S.E. wrote the first draft and jointly wrote subsequent drafts of the manuscript with K.Y.C., I.J.W., M.D.K. and L.T.V. All co-authors commented on versions of the manuscript.

## Competing interests

The authors declare no competing interests.

## Additional information

**Supplementary information** The online version contains

supplementary material available at https://doi.org/10.1038/s41467-022-32545-0.

[1]Hawkesbury Institute for the Environment, Western Sydney University, Penrith, NSW, Australia. [2]School of Biological Sciences, University of Bristol, Bristol, UK. [3]ARC Centre of Excellence for Climate Extremes, University of New South Wales, Sydney, NSW, Australia. [4]Department of Biology, University of Antwerp, Antwerp, Belgium. [5]Laboratoire des Sciences du Climat et de l'Environnement (LSCE), Institut Pierre Simon Laplace, CEA/CNRS/Université de Versailles Saint-Quentin-en-Yvelines/ Université de Paris Saclay, Gif-sur-Yvette, France. [6]Lehrstuhl für Physische Geographie mit Schwerpunkt Klimaforschung, Universität Augsburg, Augsburg, Germany. [7]Max Planck Institute for Biogeochemistry, Jena, Germany. [8]Department of Life Sciences, Imperial College, London, UK. [9]Centre for Tropical Environmental and Sustainability Science, College of Science and Engineering, James Cook University, Cairns, Australia. [10]Faculdade de Filosofia, Ciências e Letras de Ribeirão Preto, Depto. de Biologia, Universidade de São Paulo-Ribeirão Preto, Ribeirão Preto, Brazil. [11]Department of Biological and Environmental Sciences, University of Gothenburg, Gothenburg, Sweden. [12]College of Life and Environmental Sciences, University of Exeter, Exeter, UK.

[13]National Institute of Amazonian Research (INPA), Manaus, Brazil. [14]Department of Agricultural and Food Sciences, University of Bologna, Bologna, Italy. [15]Japan International Research Centre for Agricultural Sciences, Tsukuba, Japan. [16]Faculty of Agriculture and Marine Science, Kochi University, Kochi, Japan. [17]Research School of Biology, The Australian National University, Canberra, ACT, Australia. [18]School of Geosciences, Edinburgh University, Edinburgh, Scotland, UK. [19]Department of Ecology and Evolutionary Biology, University of Tennessee, Knoxville, TN, USA. [20]Department of Forest Resources, University of Minnesota, St. Paul, MN, USA. [21]Institute for Global Change Biology, and School for the Environment and Sustainability, University of Michigan, Ann Arbor, MI 48109, US. [22]Department of Botany and Plant Sciences, University of California, Riverside, Riverside, CA, USA. [23]College of Marine Life Sciences, Ocean University of China, Qingdao, China. [24]Environmental Sciences Division and Climate Change Science Institute, Oak Ridge National Laboratory, Oak Ridge, TN, USA. [25]Department of Crop Science, Faculty of Agriculture, University of Peradeniya, Peradeniya, Sri Lanka. [26]CAS Key Laboratory of Tropical Forest Ecology, Xishuangbanna Tropical Botanical Garden, Chinese Academy of Sciences, Mengla, Yunnan, China. [27]Department of Biological Sciences, Macquarie University, North Ryde, NSW, Australia. ✉e-mail: D.Ellsworth@westernsydney.edu.au

