## [Peer Review File · Nature Communications]

Response to reviewer comments, first round –

Reviewer #1 (Remarks to the Author):

The submitted manuscript describes the influence of leaf phosphorus (P) content on photosynthesis in tropical angiosperms. As well-explained by the authors, this topic is particularly important considering the limiting role of P on vegetation. As such, I consider that this study has the potential to interest a wide audience, such as whose of Nature Communications. The manuscript is well-written and easy to follow. Above all, the study design and the findings are robust, and I recommend their publication provided that a moderate revision is carried out.

**** Introduction (lines 77-171)**

This section is fairly good. But it could be sometimes confusing for readers that are not deeply experienced in plant ecophysiology, as many readers of Nature Communications are. I recommend to be more explicit when speaking of "photosynthesis" (lines 84, 86, 90, 96, 120, 137, 145 and so on). For instance, line 86, is it a reference to A.net? Because the manuscript constantly shifts from one component of photosynthesis to another, it is important to avoid confusion.

In addition, it would be good to propose a functional scheme (as a Figure or as a Box) that explains the different component of photosynthesis (A.net, Vc.max, J.max), their unit, their process-based relationships, and how they may be controlled by nitrogen and phosphorus (through Rubisco, ATP and so on). This scheme would be very useful to illustrate the passages from line 89 to line 123.

- line 98: cite "Vcmax" here (with unit).
- line 110: give the unit.
- line 102: see the Figure 2 ("gap 1") and the Table 2 (first row) in Achat et al. (2016).
- line 128: please cite the model.
- line 167: replace "the key leaf" by "a key leaf".

**** Results (lines 172-265)**

One important result (according to me) is that the influence of leaf P is at least as strong as the influence of leaf N (Supplementary Table 4). Therefore, I recommend to merge the Table S4 with the Table 1, forming a revised Table 1.

I am not fully convinced by the Figure 3 (and related passage at lines 236-243). Indeed, as acknowledged by the authors (line 347), the difference in slope value is small. Perhaps I missed the information, but it does not seem that this difference is statistically significant (please clarify). In addition, the grouping in two N:P classes (low and high; 12 and 35) is confusing because all species of a class do not have the same N:P value. The classes are likely rather defined by a single threshold value (as for leaf P content). Neither the figure captions nor the Methods section are helpful to understand this data analysis. I recommend to (at least) clarify the explanations, and maybe to revise the data interpretation (if the slopes are not significantly different).

- lines 219-221: see below the minor comment about line 64.
- line 229: it would be good to mention the direction (i.e. positive or negative) of Ma's influence. It is indicated in the Supplementary Note 1, but please indicate this information also in the main text.

**** Discussion (lines 266-316)**

This section is really interesting and pleasant to read. I do not have any major criticism to address, but have one regret. Indeed, while reading the introduction section (149-158), I thought to myself to what extent these findings would be applicable to temperate and boreal forests, to gymnosperms, and even to non-spermatophytes. In a later stage of my reading, the Supplementary Figure 6 gave a partial answer to this question. According to me, it is a pity that only a very few non-tropical species were used in this part of the data analyses. It took me only a short period of search to find relevant studies (Porté & Loustau 1998; Delzon et al. 2005; Bown et al. 2007). See in particular Delzon et al. (2005). In my opinion, even publications without values for leaf P content (e.g. Dreyer et al., 2001) can be used, provided that the studied tree species is common enough (e.g. *Picea abies*, *Fagus sylvatica*, or *Acer saccharum*) to have leaf P values in databases such as TRY. All in all, I strongly encourage the authors to enlarge their dataset about non-tropical

tree species because it would substantially strengthen the potential impact of their findings.

** Functional balance of the biochemistry of photosynthesis (lines 318-351): see the comment about the Figure 3 and the N:P ratio.

** Model analysis and implications (lines 353-403)

This section is also interesting. But I think that some claims should be toned down. Notably, it is not new that incorporating P in global models has for main consequence to decrease the ecosystems carbon sink. See the Figure 1D in Achat et al. (2016) and the cited references therein. This section is also a bit too long and would benefit from a shortening, without losing information or impact.

** Methods

- lines 429: I suggest to expend this dataset (see above)
- lines 443-446 (and lines 174-177): Low leaf P does not systematically correspond to cases of low P availability. Indeed, it can be only a species-specific adaptation to P scarcity, this species being in conditions of adequate P supply, and vice versa (see Turner et al., 2008). Because of the Supplementary Figure 4, it seems that there is no bias in this study, but I think this possible drawback should be mentioned.

** Other (minor) comments:

- line 64: the reference 3 does not support this statement. It should be moved to line 62. Instead, rather cite Hou et al. (2020, 2021) and/or reference 31.
- lines 66-67: cite a reference (for instance Achat et al. (2016)).
- line 103-104: this is correct, but the cited references are not adequate. Consider cite Vitousek et al. (2010) instead of reference 10. In addition, see above the comment about line 64.

** References:

Achat, D. L., Augusto, L., Gallet-Budynek, A., & Loustau, D. (2016). Future challenges in coupled C–N–P cycle models for terrestrial ecosystems under global change: a review. *Biogeochemistry*, 131(1-2), 173-202.

Bown, H. E., Watt, M. S., Clinton, P. W., Mason, E. G., & Richardson, B. (2007). Partitioning concurrent influences of nitrogen and phosphorus supply on photosynthetic model parameters of *Pinus radiata*. *Tree Physiology*, 27(3), 335-344.

Delzon, S., Bosc, A., Cantet, L., & Loustau, D. (2005). Variation of the photosynthetic capacity across a chronosequence of maritime pine correlates with needle phosphorus concentration. *Annals of Forest Science*, 62(6), 537-543.

Dreyer, E., Le Roux, X., Montpied, P., Daudet, F. A., & Masson, F. (2001). Temperature response of leaf photosynthetic capacity in seedlings from seven temperate tree species. *Tree physiology*, 21(4), 223-232.

Hou, E., Luo, Y., Kuang, Y., Chen, C., Lu, X., Jiang, L., ... & Wen, D. (2020). Global meta-analysis shows pervasive phosphorus limitation of aboveground plant production in natural terrestrial ecosystems. *Nature communications*, 11(1), 1-9.

Hou, E., Wen, D., Jiang, L., Luo, X., Kuang, Y., Lu, X., ... & Luo, Y. (2021). Latitudinal patterns of terrestrial phosphorus limitation over the globe. *Ecology Letters*.

Porté, A., & Loustau, D. (1998). Variability of the photosynthetic characteristics of mature needles within the crown of a 25-year-old *Pinus pinaster*. *Tree physiology*, 18(4), 223-232.

Turner, B. L., Brenes-Arguedas, T., & Condit, R. (2018). Pervasive phosphorus limitation of tree species but not communities in tropical forests. *Nature*, 555(7696), 367-370.

Vitousek, P. M., Porder, S., Houlton, B. Z., & Chadwick, O. A. (2010). Terrestrial phosphorus limitation: mechanisms, implications, and nitrogen–phosphorus interactions. *Ecological applications*, 20(1), 5-15.

Reviewer #2 (Remarks to the Author):

The key message of this study is that photosynthesis is limited by soil phosphorus (P) availability for most subtropical and tropical forests and therefore all terrestrial biosphere models (TBMs)

should account for P limitation of photosynthesis in addition to the commonly used N limitation. Potential gross photosynthesis estimated in the absence of P limitation in the ORCHIDEE-CNP model was greatly overestimated compared to estimates of photosynthesis accounting for P constraints on photosynthesis in this model, which agrees with empirical up-scaling of eddy covariance measurements and MODIS-estimated GPP. The manuscript is overall clearly written and pleasant to read.

The modeling approach used here is valid for demonstrating the limitation of photosynthesis by phosphorus in tropical and subtropical forests, despite the lack of experimental demonstration in response to P addition (in fertilization experiments for example). More than 150 species covering a wide range of leaf N and P concentrations sampled from 4 continents (America, Africa, Asia and Australia) were assembled for the first time, providing a reliable insight into the response of the relationship between leaf N and P concentrations and photosynthetic parameters (V_{cmax} and J_{max}).

The significance of the results to improve current global terrestrial models is therefore high. This paper could help change the current belief that net primary productivity of natural terrestrial ecosystems is primarily limited by nitrogen, by confirming that there is often a multi-elemental limitation, including phosphorus deficiency that is common in tropical and subtropical regions.

The manuscript references previous literature appropriately. It may be worth adding in the Discussion section that further work is needed to disentangle the effects of changes in the botanical composition of natural vegetation in response to widely varying soil N and P availability from the response of individual species to contrasting N and P supply. In addition, it is unlikely that most TBMs currently in use have overestimated the GPP by about half compared to the empirical scaling of eddy covariance measurements and atmospheric inversion models. Can you add a brief discussion to explain why most TBMs have been considered accurate enough so far without considering a limitation due to low foliar P contents?

Below are the line numbers of the sentences that need to be clarified.

Title: your study focuses on tropical and subtropical regions and the title suggests a convergence of phosphorus constraints in photosynthesis in all forests of the world. You should add "tropical and subtropical forests" in the title, because the 5 sites for *Pinus* and *Quercus* species in the northern hemisphere in Supplementary Figure 6 are not sufficient to generalize to all forests of the world.

L70-73: 36% seems huge in the abstract and it would be surprising if current global terrestrial carbon models overestimated gross photosynthesis by that much. This issue is satisfactorily addressed in the Discussion section and this sentence could be modified in the abstract to not suggest that all photosynthetic values currently considered in global C models are so overestimated for tropical and subtropical forests.

L178: are we talking about species or plants here? Do you confirm that, depending on the site, no plants of the same species were found in both the "moderate-P" and "low-P" classes?

L201: 'more specifically' could be deleted. Leaf P effects on V_{cmax} were also highly significant in Fig. 1.

L231-234: should be checked in Supplementary table 3, not significant for A_{net} and <0.02 for V_{cmax} and J_{max} .

L241-243: this is not obvious in Fig. 1 and not shown with a statistical test in Fig. 3. V_{cmax} is not shown in Fig. 2.

L253: 'equatorial zone' should be replaced by 'tropical and subtropical zone'.

L254-265: Does this suggest that the default assumption of the CMIP6 models leads to an

overestimation of about half of the total annual global C uptake for tropical regions? The discrepancy seems too large not to have been accounted for when comparing the predicted values with the empirical scaling of eddy covariance measurements. This section should be rewritten to mitigate the potential error in the global C models currently used.

L269: 'the basis of expression' should be clarified (P-values for slope differences are not significant in Suppl. Table 3 for area-based Anet, Vcmax and Jmax).

L281: ...at low leaf Pmass (and moderate or high leaf Nmass), ...

L283: taxa are not shown in Fig. 2.

L325: (Fig. 1). The role of P in regulating Vcmax is not clear in Fig. 3.

L324-328: you need to be more careful here because you are comparing many species adapted to different soil types. The response of Vcmax and Jmax is not as direct as if you had measured the values in the leaves of the same species in response to P supply.

L403: I recently reviewed a manuscript showing that K availability commonly limits tree growth in natural forests around the world and K availability can also significantly influence photosynthesis. You might add a sentence in conclusion to indicate that multi-element limitation is common and that further studies are needed to explore the need to consider other nutrients (beyond N and P) that may control C3 plant photosynthesis in TBMs.

L446: you might add here mean and std values for each leaf P class: 'moderate P' and 'low P'.

L495-496: ... 'the prognostic Ma predicted in the model': can you elaborate?

Fig. 1. The sentence 'different symbols... combinations' could be deleted. The symbols are difficult to distinguish in the figure. Why are there different colors in (c) and (d)?

Fig. 3. The regression for N:P=12 is driven by 2 points with N:P values > 200 and the variability is high. Can you show using a statistical test that the slopes are significantly different between the two regressions?

Supplementary Figure 1. (c,d) Vc should be replaced by J.

L120: Fig. 1b,d should be replaced by Fig. 1c,d.

Supplementary Figure 3. It might be interesting to indicate whether there are any N-fixing species in your dataset and where they are located in this phylogenetic map. N-fixing species generally have high leaf N concentrations and high P requirements for symbiotic fixation. Can you show that the slopes in Figure 1 are not due to an over-representation of N-fixing species in your dataset compared to tropical forests?

Supplementary Figure 4. a), b), c), d) must be included in the table title.

Supplementary Figure 5. The symbols are difficult to read (except the color). It would be better to indicate the meaning of the symbols here.

Supplementary Figure 7. (a) and (b) should be added in the Figure.

Supplementary Figure 8. It should be specified that the ORCHIDEE-CNP model has been used.

Supplementary Table 1. L204: ... includes the country, the elevation at the study site, ...

Jean-Paul Laclau

Reviewer #3 (Remarks to the Author):

This paper is a very nice contribution to our understanding on how chronic status of P influences photosynthesis biochemistry in the tropics and the demonstration of the use of the general relationships in ORCHIDEE-CNP is convincing. I congratulate the authors to have done such a great work. The paper should be accepted for publication and I believe it is of wide interest. Below are of my minor comments, which I hope can help the authors to clarify some points:

About the method section: There are several comparisons for slopes of different groups in the paper. How the significance of the difference in slopes between/among different groups is tested? No such information is provided in the method section.

line 61: needs to say this absorption of 35 Pg C is gross photosynthesis, avoiding that people might think this is carbon sink.

Line 118 of Supplementary information: the latter V_{cmax_mass} should be J_{max_mass}

line 64: it is more of this general quantitative relationship reported in this paper is lacking. Might deserve some stress in the abstract.

line 73-75: this study also provides a quantitative relationship that could be used by models.

Perhaps worth mentioning, but not only stressing the importance of P limitation per se.

line 195: Fig. 1c and 1d do not indicate the r^2 for regressions. Authors need to provide evidence that 'leaf P_{mass} on its own explains more variation in maximum ... than did leaf N on its own'.

line 207-210: There are two arguments here. But I think their supporting evidences are swapped? Supplement Table 5 should be used for line 207, while Fig. 2 should be used for line 208-209? It is a nice way to show Fig. 2 in its current form but the mixing of the other three continents can blur the comparison

when looking at a specific continent. I would think putting Supplement Table 5 in the main text is better. This is the key information. While Fig. 2 looks nice, but its information is more of a complementary role. But if the authors can justify why Fig. 2 is more appropriate, I would accept their decision.

Supplementary Fig. 5: Should we look at the relationship between the residual of regressions of $V_{cmax} \sim N+P$ and climate/aridity variables?

line 221: The authors did not *analyze* the dataset from Europe and North America in the same manner as the large dataset covering tropical continents.

The datasets are just plotted together. This only shows that datasets from northern hemisphere are not outliers.

line 231-234: I get confused when looking at Supplementary Table 3. The intercept for Anet is not significant, while for V_{cmax} and J_{max} P-value is bigger than 0.005... The last two lines of Supplementary Table 3 has both as "Mod. P", pls correct.

Fig. 3 is nice. But how lowN/P and high N/P are defined? Please double check the caption.

line 241-243: How do Fig. 1 and Fig. 2 show 'stronger effects of leaf P on J_{max} '. I am confused.

line 282-287: related to my previous comment, Fig. 2 is a less convincing way to show this than Supplementary Table 5.

“Convergence in phosphorus constraints to photosynthesis in forests around the world” by Ellsworth et al. – Responses to Reviewers Comments

Responses to the Reviewers appear in blue text.

Reviewer #1

**** Introduction (lines 77-171)**

This section is fairly good. But it could be sometimes confusing for readers that are not deeply experienced in plant ecophysiology, as many readers of Nature Communications are. I recommend to be more explicit when speaking of “photosynthesis” (lines 84, 86, 90, 96, 120, 137, 145 and so on). For instance, line 86, is it a reference to A.net? Because the manuscript constantly shifts from one component of photosynthesis to another, it is important to avoid confusion.

We apologise for the lack of clarity and we have revised the text. ‘Photosynthesis’ is a physiological process and should be referred to generically when the overall process is discussed, but the Reviewer is correct that when discussed in specific we should state whether it is ‘net’ or ‘gross’ photosynthesis under consideration. In the text we clarified whether we refer to net photosynthesis or gross photosynthesis, in keeping with the Reviewer’s comments. Photosynthetic capacity is an attribute that applies equally to gross and net photosynthesis, so we have left that as-is. When we refer to underlying components of photosynthesis, we have been clear about that and which component is under consideration, as the Reviewer has pointed out.

In addition, it would be good to propose a functional scheme (as a Figure or as a Box) that explains the different component of photosynthesis (A.net, Vc.max, J.max), their unit, their process-based relationships, and how they may be controlled by nitrogen and phosphorus (through Rubisco, ATP and so on). This scheme would be very useful to illustrate the passages from line 89 to line 123.

We agree and we want to increase accessibility to the broader scientific community. We have added a Figure for this purpose, which we have included in the Supplementary Material (new Supplementary Figure 1). Nature Communications does not have a box-format.

- line 98: cite “Vcmax” here (with unit). Done.
- line 110: give the unit. Done.
- line 102: see the Figure 2 (“gap 1”) and the Table 2 (first row) in Achat et al. (2016). Done.
We added: ‘including those central to the terrestrial C cycle^{20,21}.’ (l. 45 in the revision).
- line 128: please cite the model. The model is CABLE-CASA-CNP (Wang et al. 2010), and this has been cited where the Reviewer indicated (l. 73-74).
- line 167: replace “the key leaf” by “a key leaf”. Done.

**** Results (lines 172-265)**

One important result (according to me) is that the influence of leaf P is at least as strong as the influence of leaf N (Supplementary Table 4). Therefore, I recommend to merge the Table S4 with the Table 1, forming a revised Table 1.

We find this suggestion very useful and we have done so in the revision. Table S4 has been removed.

I am not fully convinced by the Figure 3 (and related passage at lines 236-243). Indeed, as acknowledged by the authors (line 347), the difference in slope value is small. Perhaps I missed the information, but it does not seem that this difference is statistically significant (please clarify).

We apologise that we have not fully presented the supporting analyses for Fig. 3 in the previous version. We have generated a new Figure 3 and supporting analyses which is more direct in using leaf P concentrations rather than N:P, while making a similar point to the original analysis. The new Figure 3:

The difference between the two P_{mass} classes that we show above was statistically significant at $P = 0.035$ (stated in the figure legend of the revised Fig. 3). This difference was tested by a separate slopes analysis using P_{mass} class as a covariate in interaction with the independent variable ($V_{c\max}$), which has been clarified according to Reviewer #3's request (new Fig. 3 caption). The new Figure 3 we have generated is a more direct analysis than our previous Fig. 3, but makes a very similar point to the original analysis.

Text has been added to the Figure 3 legend for how the differences between leaf P_{mass} classes were tested (new Fig. 3 caption) and in the methods section (l. 469-477).

In addition, the grouping in two N:P classes (low and high; 12 and 35) is confusing because all species of a class do not have the same N:P value. The classes are likely rather defined by a single threshold value (as for leaf P content).

We did state previously that these N:P groups are classes. However, we have reoriented the analysis towards using mean leaf Pmass classes. We have clarified the mean low and high Pmass classes including the standard deviation (s.d.) within each class in the Figure 3 caption text:

“The lines shown are for the two end-member leaf Pmass classes: mean low Pmass of 0.44 ± 0.11 (s.d.), and mean high Pmass of 1.76 ± 0.55 (s.d.)”.

Neither the figure captions nor the Methods section are helpful to understand this data analysis. I recommend to (at least) clarify the explanations, and maybe to revise the data interpretation (if the slopes are not significantly different).

We agree that we have not completely described this analysis in the original submission. This comment is partly obviated by the new Figure 3. The slopes of the Pmass classes in revised Figure 3 were significantly different at $P = 0.035$, which we now report in the new Figure 3 legend. We also state how we tested this, in the legend and also l. 469-477 in the revised version.

- lines 219-221: see below the minor comment about line 64. We now have cited Hou et al. (Ref #43) here (l. 174 in the current revision).

- line 229: it would be good to mention the direction (i.e. positive or negative) of Ma's influence. It is indicated in the Supplementary Note 1, but please indicate this information also in the main text.

We inserted 'with reduced slope' in the main text as advised by the Reviewer (revised version l. 185).

** Discussion (lines 266-316)

This section is really interesting and pleasant to read. I do not have any major criticism to address, but have one regret. Indeed, while reading the introduction section (149-158), I thought to myself to what extent these findings would be applicable to temperate and boreal forests, to gymnosperms, and even to non-spermaphytes. In a later stage of my reading, the Supplementary Figure 6 gave a partial answer to this question. According to me, it is a pity that only a very few non-tropical species were used in this part of the data analyses. ... All in all, I strongly encourage the authors to enlarge their dataset about non-tropical tree species because it would substantially strengthen the potential impact of their findings.

Thank you for these comments. An extensive literature review and data acquisition exercise to satisfy the Reviewer's regret is outside the scope of this effort, but we sought to address this in a different way. We undertook a compilation of data from the TRY database to address this comment. P data from temperate and boreal regions are rare in TRY which constitutes the major leaf trait database, enhancing the value of the data we contributed in the original Supplemental Figure 6, which increases the temperate database in TRY for species-at-site by 3-fold at least.

Specifically, to address this comment we requested information from the TRY database with 4.54 million observations in April 2022. In TRY, there were 182,025 relevant observations that included one of the following: V_{cmax} , J_{max} , leaf N concentration, leaf P concentration. **There is only one single temperate zone observation in all of TRY where V_{cmax} and P were measured at the same species-site combination at a temperate site, e.g. in a manner like the data we specified on l. 164 of the original submission.**

We relaxed assumptions in accord with the Reviewer suggestion to advance this analysis further to test if temperate zone observations agree with our other continents (new Supplemental Figure 6, now renumbered Supplemental Figure S7):

Grouped according to temperate sites and species in a relaxed analysis (e.g., allowing species-level but not species-at-site means), only 38 observations from temperate species (including 6 gymnosperm species) had any leaf P concentration data at all, alongside V_{cmax} and/or J_{max} and M_a (only 12 such species in TRY for J_{max}). This illustrates how rare the data in the manuscript are, against one well-known data archive.

To account for the reviewers' comment, we have revised Supplementary Figure 6 (now Supplementary Figure 7) to show the temperate data that we provided as well as data from the TRY database. We must point out that this is not a particularly strong analysis, and we abstain from drawing a slope through the data in Supplementary Figure 7 because 1) they have not been temperature-corrected like the remainder of the data in both Figure 1 and the former Supplementary Figure 6 were, and 2) they are not observations from species-at-site (original version line 164), much less from the same plants that were measured, and hence are not consistent with the original data we presented in the first version of the manuscript. Had we used the native temperature corrected of either V_{cmax} or J_{max} , which is the standard for presenting these parameters after Leuning 2002, then we would have had even less of this data to add from TRY.

The point is that **few studies measuring V_{max} or J_{max} in gas exchange in the temperate zone actually also measure leaf P**. P is generally not considered limiting photosynthesis in the temperate zone but more such data are needed to generalise across all 6 vegetated continents. We discuss this in text l. 254-261 in our revisions:

“There is evidence that relationships like J_{max}_mass–P_{mass} are generalizable to Northern Hemisphere temperate woody plants (Ref. 19,49 and Supplementary Figure 7) but in this regard there is a clear need for further work involving temperate coniferous and deciduous trees. In fact, in the extensive TRY database, there is a paucity of Northern Hemisphere temperate records (Supplementary Figure 7), particularly involving species-at-site values for J_{max}, Ma and P_{mass}. We identify this as an area for further research, involving both broadleaved and needle-leaved temperate and boreal species.”

It took me only a short period of search to find relevant studies (Porté & Loustau 1998; Delzon et al. 2005; Bown et al. 2007). See in particular Delzon et al. (2005). In my opinion, even publications without values for leaf P content (e.g. Dreyer et al., 2001) can be used, provided that the studied tree species is common enough (e.g. *Picea abies*, *Fagus sylvatica*, or *Acer saccharum*) to have leaf P values in databases such as TRY.

Please see our response to part of this comment above. As for ‘relevant studies’, we did employ 4 data requirements in our manuscript that are not satisfied by the Reviewer’s brief data search. These are: 1) mature life stages are needed (not seedlings) see line 413 of the original submission, 2) similar instrumentation and data format (line 408 of the original version), 3) raw data with supporting measurements was needed (line 160 and l. 459-471 of original version) and 4) leaf P concentration data were available from the same plants (originally, lines 421-422). Each of the studies cited above fails on at least 2 of these approaches, or one approach if we relax #2. Thus, these studies suggested by the Reviewer still cannot be incorporated in the dataset. We have detailed the selection criteria for data in Supplemental Figure 10.

We also cannot take leaf P concentrations measured from other locations than the ones where photosynthesis was measured using species information though this is suggested by the reviewer. Many recent studies have highlighted the importance of site (Yan et al. 2019, ref 34 and Hidaka and Kitayama 2009), soil type (Yang & Post 2011, ref 36) and parent material (Augusto et al. 2016, Crous & Ellsworth 2020, He et al. 2022) influencing leaf P, and one cannot ignore these and take leaf P concentration values from the same species growing elsewhere. An entirely new search for data that is not following our criteria for data is not possible at this time or in the frame of this revision, but it is a good idea for another manuscript.

** Functional balance of the biochemistry of photosynthesis (lines 318-351): see the comment about the Figure 3 and the N:P ratio.

We apologise that the description of Figure 3 was incomplete and we have rectified the problem in the revision of Figure 3. The new revised text is l. 318-323: “The J_{max}/V_{max} shift with increasing P_{mass} in our dataset is not large (e.g., a 10% reduction with low P_{mass};

Figure 3), supporting a functional balance for the components of the photosynthetic apparatus. Still, the majority of TBMs that parameterise J_{max} based on the basis of this functional balance with V_{cmax} and a highly conserved J_{max}/V_{cmax} should consider these changes in the ratio with low leaf P_{mass} ".

**** Model analysis and implications (lines 353-403)**

This section is also interesting. But I think that some claims should be toned down. Notably, it is not new that incorporating P in global models has for main consequence to decrease the ecosystems carbon sink. See the Figure 1D in Achat et al. (2016) and the cited references therein. This section is also a bit too long and would benefit from a shortening, without losing information or impact.

We agree that a number of models have incorporated aspects of P biogeochemistry as a constraint, and Goll et al. (2012) as well as Wang et al. (2010), Zhang et al. (2014), Weider et al. (2015), Yang et al. (2019), Wang et al. (2020) are all examples of this (Refs at the bottom of our responses). This is an emerging research area that our manuscript contributes to (Achat et al. 2016, Fleischer et al. 2017, Reed et al. 2016, Wang & Goll 2021). However, in spite of the outcome having qualitatively similar results in our paper as these previous papers, the Reviewer has missed a key part of this constraint that is in fact new in this manuscript: **none of these manuscripts nor the models in them had P limitations on the incoming flux of CO₂ dependent on the biochemistry of photosynthesis** (l. 394-395 in the original manuscript). This aspect has previously been unknown or believed to be site dependent and not robust enough for inclusion in models (see Reed et al 2016). See text in our original submission, l. 276-277, l. 389-390. From Reed et al. (2016): "unique challenges for P include foliar concentrations that are much more variable than N, and our poor understanding of how leaf P concentrations control photosynthesis. As a result, a mechanistic representation of leaf P controls on photosynthesis have not been implemented in models."

References are at the end of our response letter.

**** Methods**

- lines 429: I suggest to expend [sic] this dataset (see above) We have done so in the new Supplementary Figure 8, also described l. 405-410 in the revised text:

"A small, limited dataset from Europe and North America that was compiled from direct measurements and from the TRY database⁴⁴ analysed in the same manner as our large and diverse cross-continent dataset (Supplemental Figure 8) to compare the results with deciduous and gymnosperm species (5 species of *Quercus* and *Pinus*, as major Northern Hemisphere genera)."

- lines 443-446 (and lines 174-177): Low leaf P does not systematically correspond to cases of low P availability. Indeed, it can be only a species-specific adaptation to P scarcity, this species being in conditions of adequate P supply, and vice versa (see Turner et al., 2008). Because of the Supplementary Figure 4, it seems that there is no bias in this study, but I think this possible drawback should be mentioned. We agree and have revised the text to state this potential drawback as suggested: "In so doing, we recognize that low leaf P

concentrations may not always reflect soil availability due to species-level mechanisms that can affect P uptake⁵⁸." (l. 423-426).

**** Other (minor) comments:**

- line 64: the reference 3 does not support this statement. It should be moved to line 62. Instead, rather cite Hou et al. (2020, 2021) and/or reference 31.

The journal does not permit references in the abstract, so this particular reference has been removed from here.

- lines 66-67: cite a reference (for instance Achat et al. (2016)). The journal does not permit references in the abstract, so I'm unable to satisfy the Reviewer in this regard, though we have cited Achat et al. (2016) elsewhere (Ref 20) in accord with the comment.

- line 103-104: this is correct, but the cited references are not adequate. Consider cite Vitousek et al. (2010) instead of reference 10. In addition, see above the comment about line 64.

The Reviewer is correct and we have cited the preferred reference as suggested.

**** References**

We have incorporated several of these references as relevant.

Reviewer #2 (Remarks to the Author):

The key message of this study is that photosynthesis is limited by soil phosphorus (P) availability for most subtropical and tropical forests and therefore all terrestrial biosphere models (TBMs) should account for P limitation of photosynthesis in addition to the commonly used N limitation. Potential gross photosynthesis estimated in the absence of P limitation in the ORCHIDEE-CNP model was greatly overestimated compared to estimates of photosynthesis accounting for P constraints on photosynthesis in this model, which agrees with empirical up-scaling of eddy covariance measurements and MODIS-estimated GPP. The manuscript is overall clearly written and pleasant to read.

The modeling approach used here is valid for demonstrating the limitation of photosynthesis by phosphorus in tropical and subtropical forests, despite the lack of experimental demonstration in response to P addition (in fertilization experiments for example). More than 150 species covering a wide range of leaf N and P concentrations sampled from 4 continents (America, Africa, Asia and Australia) were assembled for the first time, providing a reliable insight into the response of the relationship between leaf N and P concentrations and photosynthetic parameters (V_{cmax} and J_{max}).

The significance of the results to improve current global terrestrial models is therefore high. This paper could help change the current belief that net primary productivity of natural terrestrial ecosystems is primarily limited by nitrogen, by confirming that there is often a multi-elemental limitation, including phosphorus deficiency that is common in tropical and subtropical regions.

We appreciate the valuable comments of the reviewer, and we thank him for understanding our scientific contribution in this manuscript.

The manuscript references previous literature appropriately. It may be worth adding in the Discussion section that further work is needed to disentangle the effects of changes in the botanical composition of natural vegetation in response to widely varying soil N and P availability from the response of individual species to contrasting N and P supply.

We have added a brief comment to this effect, l. 243-246 (revised version): “Further work is needed to disentangle changes in the botanical composition of natural vegetation in response to varying soil N and P availability from the response of individual species to contrasting N and P supply”.

In addition, it is unlikely that most TBMs currently in use have overestimated the GPP by about half compared to the empirical scaling of eddy covariance measurements and atmospheric inversion models. Can you add a brief discussion to explain why most TBMs have been considered accurate enough so far without considering a limitation due to low foliar P contents?

The reviewer raises an important point about TBMs. Unfortunately, it is an accurate reflection of the uncertainty in our capacity to simulate tropical GPP. An overestimation of up to a half could occur. For example, Seiler et al. (2022) showed that the TRENDY models simulate a range in GPP for the tropics from ~5.5 to 10 g C m⁻² d⁻¹ (FLUXCOM estimate ~7.7 g C m⁻² d⁻¹). Please note there is a lack of eddy covariance sites in the tropics that underpin this observational estimate, leaving large uncertainties in GPP estimates.

Model tuning of parameters controlling the photosynthetic capacity has been a practice within the land surface model community (Goll pers. comm.), given that they are annually benchmarked against each other and various data products (e.g. as part of the Global Carbon Project), implying the uncertainty is likely to be at least within the range of observation-based estimates. Looking at the estimates from the CMIP6 ensemble, Hu et al. 2022 showed that the tropical GPP ranged between ~4400 and ~1500 g C m⁻² y⁻¹ (although note the lower range is more like ~2400 g C m⁻² y⁻¹ for most of the models, the ensemble has one extreme outlier).

A large number of TBMs deploy plant functional type (PFT) specific parameters for photosynthesis (i.e. J_{\max} and V_{\max}) and thus, if parameterized on present-day measurements, can implicitly capture the large scale variation in photosynthetic parameters (which we argue are driven by N and P) simply by the use of maps of the distribution of PFTs derived from earth observation. It has to be noted that large variation within the tropical PFTs was observed, which could be related to differences in soil types (Kattge et al. 2007) - questioning this approach and might explain the large model spread regarding tropical GPP shown in Seiler et al (2022) (l. 340 in the revised text).

The tuning of V_{\max} and J_{\max} has been reduced by the introduction of N cycles in LSMs, which often deploy an empirical relationship of V_{\max} and J_{\max} with leaf N (nitrogen use efficiency (NUE); Kattge et al. 2007) rather than prescribing J_{\max} and V_{\max} . However, due to the large variation in this slope for tropical plant functional type (PFTs) (i.e. Kattge et al. 2007), it

opened the door again for tuning of NUE of tropical PFTs in models with a P cycle in addition to N (e.g. Goll et al 2017). Our new formulation now also allows taking leaf P as an additional constraint into account, which we argue will allow more robust model predictions.

A major shortcoming of approaches omitting P effects, is that the effect of changing phosphorus availability (see Peñuelas et al 2013 Nature Commun, or Jonard et al 2015) on photosynthesis cannot be captured, as this requires to resolve the link between Anet and leaf phosphorus concentration implicitly. As a consequence, they might overestimate the effect of increasing CO₂ on GPP, or underestimate the effect of changing atmospheric nutrient deposition in the past and future.

Our major message is not that the TBMs are inaccurate in capturing the few uncertain available observations for recent decades, but uncertainties can be improved and they can incorporate correct physiological mechanisms, enhancing our predictive skills. Our original version provided a caveat of the 'downstream processes' in models l. 369-371 that ultimately affect the outcomes in terms of C cycling and NPP, and this text is retained in the current version (now l. 340-345).

Below are the line numbers of the sentences that need to be clarified.

Title: your study focuses on tropical and subtropical regions and the title suggests a convergence of phosphorus constraints in photosynthesis in all forests of the world. You should add "tropical and subtropical forests" in the title, because the 5 sites for Pinus and Quercus species in the northern hemisphere in Supplementary Figure 6 are not sufficient to generalize to all forests of the world. We respect the Reviewer's suggestion, but it is at odds with Reviewer #1 and the further analysis s(he) suggests involving strengthening representation by temperate and boreal species. We have left the title as-is, because this further analysis (see Supplemental Figure 8) supports that the relationships like Fig. 1 and Table 1 are likely general amongst broadleaved species across all 6 vegetation continents, including temperate and boreal species where leaf P data alongside V_{cmax} and/or J_{max} are quantified. We have also highlighted this as a potential area for future research (l. 254-261):

"...but in this regard there is a clear need for further work involving temperate coniferous and deciduous trees. In fact, in the extensive TRY database, there is a paucity of Northern Hemisphere temperate records (Supplementary Figure 7), particularly involving species-at-site values for J_{max}, Ma and P_{mass}. We identify this as an area for further research, involving both broadleaved and needle-leaved temperate and boreal species".

L70-73: 36% seems huge in the abstract and it would be surprising if current global terrestrial carbon models overestimated gross photosynthesis by that much. This issue is satisfactorily addressed in the Discussion section and this sentence could be modified in the abstract to not suggest that all photosynthetic values currently considered in global C models are so overestimated for tropical and subtropical forests. Space considerations precluded mention of this in the abstract, as we needed to shorten it by 16 words rather than add words. However we were able to incorporate the caveats elsewhere in the text (l.

361-367 in the revision).

L178: are we talking about species or plants here? Do you confirm that, depending on the site, no plants of the same species were found in both the "moderate-P" and "low-P" classes? We examined the plant-level data again, and given that this is an arbitrary cut-off, yes there are plants of some species occurring in both the "moderate-P" and "low-P" classes. However, as we stated the analyses and data presentation are based on species means. So we have not changed the text here, because we made the point of using species means rather than plant-level data (Supplemental Figure 11 and l. 110 and l. 398-404 in the revision).

L201: 'more specifically' could be deleted. Leaf P effects on Vcmax were also highly significant in Fig. 1. Done.

L231-234: should be checked in Supplementary table 3, not significant for Anet and <0.02 for Vcmax and Jmax. The reviewer is correct, done.

L241-243: this is not obvious in Fig. 1 and not shown with a statistical test in Fig. 3. Vcmax is not shown in Fig. 2. Figures 1a and 1b do show effects of low P on Vcmax and Jmax. We have rectified the issue with a statistical test for Figure 3 as per comments above responding to Rev. #1's similar concerns (see revised Figure 3 and its legend which describes the test).

L253: 'equatorial zone' should be replaced by 'tropical and subtropical zone'. We specifically checked this for the equatorial zone, so if we replace by 'tropical and subtropical zone' as suggested, then we need to change the GPP figures as well. We have not made this change, though the results are substantially similar for the equatorial portion of the tropics as for the tropics on the whole.

L254-265: Does this suggest that the default assumption of the CMIP6 models leads to an overestimation of about half of the total annual global C uptake for tropical regions? The discrepancy seems too large not to have been accounted for when comparing the predicted values with the empirical scaling of eddy covariance measurements. This section should be rewritten to mitigate the potential error in the global C models currently used.

No, it does not suggest that CMIP6 models overestimate GPP for tropical regions by half, as they either tune Vcmax, Jmax or deploy parameters derived from present day observations (see reply above). We realized 'the default hypothesis among CMIP6 models' is an oversimplification and partly misleading. We thus removed it.

Nonetheless, these CMIP6 models, although simulating a reasonable present day GPP, might be subject to (positive and negative) biases in past and future changes in GPP predictions, as they do not resolve effects of changing nutrient availability (or leaf P) (like observed e.g. in Europe; Jonard et al 2015) which can be caused directly and indirectly by most of major

global change drivers (atmospheric deposition, increasing co₂, climate change, land use changes, land management).

We also rephrased the sentence in the concluding paragraph from:

“In TBMs that underlie our predictions of future carbon sink behaviour [...] likely overestimating GPP for these regions.”

to:

“In TBMs that underlie our predictions of future carbon sink behaviour [...] likely biasing GPP predictions for these regions.” (l. 371-372 in revision).

In the abstract we rephrase:

“Implementing P constraints in the ORCHIDEE-CNP model, gross photosynthesis was reduced by 36% [...] relative to [...] sufficient leaf P.”

“Implementing P constraints in the ORCHIDEE-CNP model, gross photosynthesis was reduced by 36% [...] relative to [...] unlimiting leaf P.” (l. 14 in revision)

L269: ‘the basis of expression’ should be clarified (P-values for slope differences are not significant in Suppl. Table 3 for area-based Anet, Vcmax and Jmax). The Reviewer is correct that Supplemental Table 3 shows this, but this still means a negative effect of P on photosynthesis as we stated. We have made no change here, but we agree that the Reviewer is correctly interpreting our statements.

L281: ...at low leaf P_{mass} (and moderate or high leaf N_{mass}), ... Revised to “low leaf P_{mass} in combination with moderately high leaf N_{mass}” (now l. 236-237)

L283: taxa are not shown in Fig. 2. Whilst correct that taxa are not shown in Fig. 2, the differences in taxa amongst continents is shown in Supplementary Tables 2 and 4, which we have cited in the text (now l. 239).

L325: (Fig. 1). The role of P in regulating Vcmax is not clear in Fig. 3. We agree, so this was changed to just indicate Fig. 1 here.

L324-328: you need to be more careful here because you are comparing many species adapted to different soil types. The response of Vcmax and Jmax is not as direct as if you had measured the values in the leaves of the same species in response to P supply.

We agree with the comment and we have specified ‘across species and soils’ in the text here, now l. 296 in the revised version.

L403: I recently reviewed a manuscript showing that K availability commonly limits tree growth in natural forests around the world and K availability can also significantly influence photosynthesis. You might add a sentence in conclusion to indicate that multi-element limitation is common and that further studies are needed to explore the need to consider other nutrients (beyond N and P) that may control C₃ plant photosynthesis in TBMs. We read this comment with great interest, and it would be great to know what specific manuscript is referred to. Unfortunately, the reference has not turned up in our searches and might not yet be published. We are aware of a review on K effects on photosynthesis of

a few economically valuable species (Eucalypt, Almond, hickory, olive) which concludes that current understanding is low (Cornut et al. 2020). In the absence of better evidence for our study, we believe that mention of multi-element limitations is potentially distracting from our main topic.

L446: you might add here mean and std values for each leaf P class: 'moderate P' and 'low P'.

The range for these leaf P classes is wide (hence the s.d. is large), because our threshold (0.92 mg g⁻¹ P) is close to the median P_{mass} in the dataset (e.g., n ≥ 225 in each of these classes). We did not add s.d. values for the leaf classes because this does not make sense: it is the variation in P_{mass} that is the driver of the relationships, which we referred to in the original submission (l. 158-159 in the original submission).

L495-496: ... 'the prognostic Ma predicted in the model': can you elaborate?

We mean the M_a that is predicted according to the model, in other words either assigned according to the mean for functional groups in the model, or estimated from geographical databases like in Butler et al. (2017) Ref 56.

Fig. 1. The sentence 'different symbols... combinations' could be deleted. The symbols are difficult to distinguish in the figure. Why are there different colors in (c) and (d)? We made the symbols larger in this figure. The symbols in Fig. 1c and Fig. 1d are partially transparent, so the different colours occur when there are overlapping data points.

Fig. 3. The regression for N:P=12 is driven by 2 points with N:P values > 200 and the variability is high. Can you show using a statistical test that the slopes are significantly different between the two regressions?

Part of this comment is no longer relevant given that we have reanalysed the data and reconfigures Figure 3. However, we answer the question for the old version of Figure 3 to hopefully satisfy curiosity. The Reviewer must have meant $J_{\max} > 200 \mu\text{mol m}^{-2} \text{s}^{-1}$, rather than 'N:P values > 200'. The dashed line in the figure below shows the result when the two points $J_{\max} > 200 \mu\text{mol m}^{-2} \text{s}^{-1}$ are deleted:

The slope of the dashed line for N:P ratio of 12 with putative "outliers" removed in this alternative version of Fig. 3 is still different from N:P ratio of 35 (the exact P-value for this difference is $P = 0.099$). We conclude that the slope difference between the two N:P ratio classes in the alternate figure above is not solely driven by these two putative "outliers" as suggested by the Reviewer. Since these two points in question by the Reviewer are NOT otherwise anomalous and we have no grounds to arbitrarily remove them, and since there is still a slope difference that is marginally significant when they are removed, we conclude that the original analysis is robust and not driven by possible "outliers".

At any rate, we have changed the original figure and analysis, and we have added detail of the analysis for the revised Figure 3 as requested by Reviewer #1.

Supplementary Figure 1. (c,d) V_c should be replaced by J . Done.

L120: Fig. 1b,d should be replaced by Fig. 1c,d. Done.

Supplementary Figure 3. It might be interesting to indicate whether there are any N-fixing species in your dataset and where they are located in this phylogenetic map. N-fixing species generally have high leaf N concentrations and high P requirements for symbiotic fixation. Can you show that the slopes in Figure 1 are not due to an over-representation of N-fixing species in your dataset compared to tropical forests?

Yes, we can show this. The Fabaceae is a very large family (third largest in Angiospermae based on number of known species), that is especially abundant in the tropics. We must be clear that not all Fabaceae in natural forest situations are fixing atmospheric N_2 or are even capable of nodulating (Sprent 2007, Werner et al. 2014). Also, some non-Fabaceae are N-fixing, but we can only address the Reviewer's question assuming Fabaceae are most likely to be N-fixers.

We tested for separate slopes as a function of leaf P_{mass} for Fabaceae vs. all the rest of the species, for both $V_{\text{cmax_mass}}$ and $J_{\text{max_mass}}$ (see figure below). Neither was significant ($P = 0.1925$ and $P = 0.3188$, respectively). Examining the graphs below, one can see a clear intermixture of Fabaceae (denoted 'legumes'; green points and lines) with other species (in grey):

Sprent, J. I. (2007). Evolving ideas of legume evolution and diversity: a taxonomic perspective on the occurrence of nodulation. *New Phytol.* 174, 11–25. doi: 10.1111/j.1469-8137.2007.02015.x

Werner, G. D. A., Cornwell, W. K., Sprent, J. I., Kattge, J., and Kiers, E. T. (2014). A single evolutionary innovation drives the deep evolution of symbiotic N_2 -fixation in angiosperms. *Nat. Commun.* 5:4087. doi: 10.1038/ncomms5087

Supplementary Figure 4. a), b), c), d) must be included in the table title. Done.

Supplementary Figure 5. The symbols are difficult to read (except the color). It would be better to indicate the meaning of the symbols here. Done.

Supplementary Figure 7. (a) and (b) should be added in the Figure. Done.

Supplementary Figure 8. It should be specified that the ORCHIDEE-CNP model has been used. Done.

Supplementary Table 1. L204: ... includes the country, the elevation at the study site, ... Done.

Reviewer #3 (Remarks to the Author):

This paper is a very nice contribution to our understanding on how chronic status of P influences photosynthesis biochemistry in the tropics and the demonstration of the use of the general relationships in ORCHIDEE-CNP is convincing. I congratulate the authors to have

done such a great work. The paper should be accepted for publication and I believe it is of wide interest.

We appreciate the Reviewer's interest in our work.

Below are of my minor comments, which I hope can help the authors to clarify some points:

About the method section: There are several comparisons for slopes of different groups in the paper. How the significance of the difference in slopes between/among different groups is tested? No such information is provided in the method section.

We apologise for the oversight. To compare slopes from two regression models, we first generated the results for each model individually. Then we tested for the significance of separate slopes using analysis of covariance (ANCOVA) to compare two regression lines by testing the effect of a categorical factor on a dependent variable (y-var) while controlling for the effect of a continuous co-variable (x-var). We now state (l. 461-464 in revision): "Differences in slopes in OLS regression analyses were tested using the appropriate categorical variable (e.g., continent, N:P ratio class, etc.) as a covariate in the model of interest, and determining the significance of the categorical variable in interaction with the independent variable in ANOVA" in the manuscript.

We hope this satisfactorily explains how these slope differences were tested.

line 61: needs to say this absorption of 35 Pg C is gross photosynthesis, avoiding that people might think this is carbon sink.

As in our response to Reviewer #2's similar comment, space considerations precluded mention of this in the abstract.

Line 118 of Supplementary information: the latter $V_{\text{max_mass}}$ should be $J_{\text{max_mass}}$

We agree and have made this change.

line 64: it is more of this general quantitative relationship reported in this paper is lacking. Might deserve some stress in the abstract.

We are unable to incorporate this comment because space considerations in the abstract prevent inclusion of extra wording. As it is, we needed to cut 15 words from the abstract to meet journal guidelines. We apologise for needing to do this.

line 73-75: this study also provides a quantitative relationship that could be used by models. Perhaps worth mentioning, but not only stressing the importance of P limitation per se.

Like above, we are unable to incorporate this comment because space considerations in the abstract.

line 195: Fig. 1c and 1d do not indicate the r^2 for regressions. Authors need to provide evidence that 'leaf P_{mass} on its own explains more variation in maximum ... than did leaf N on its own'.

This evidence was provided in Supplementary Table 4 which now has been folded into Table 1 as per Reviewer #2's advice.

line 207-210: There are two arguments here. But I think their supporting evidences are swapped? Supplement Table 5 should be used for line 207, while Fig. 2 should be used for line 208-209? It is a nice way to show Fig. 2 in its current form but the mixing of the other three continents can blur the comparison when looking at a specific continent. I would think putting Supplement Table 5 in the main text is better. This is the key information. While Fig. 2 looks nice, but its information is more of a complementary role. But if the authors can justify why Fig. 2 is more appropriate, I would accept their decision.

We appreciate the thoughtful comment from the Reviewer. We believe that first and foremost, the illustration of each continent in Fig. 2 is more effective than the statistics provided in Supplemental Table 5, and as a central point of the manuscript these deserve to be shown in Figure rather than Table form. As a result, we do believe that Fig. 2 is most appropriate to illustrate our findings, and that the lines from each specific continent can be visually compared in Fig. 2, both in terms of the regression line but also in terms of the range of P_{mass} in the data which is quite apparent. We also contend that the information in Supplemental Table 5 is appropriate in the Supplementary materials, and we do not wish to overload the Tables in the manuscript.

Regarding ‘mixing of the other three continents can blur the comparison’, we note with very different sample sizes for the different continents it is appropriate to show a stable cross-continent relationship for comparison, which is best done using the remaining continents in comparison. Those looking for the statistical details can always refer to Supplemental Table 4.

Supplementary Fig. 5: Should we look at the relationship between the residual of regressions of $V_{cmax} \sim N+P$ and climate/aridity variables?

We have done this analysis, but there was still no relationship between these residuals and climate. For $J_{maxmass} \sim P_{mass}$ residuals, which uses the strongest single regression, the graph is:

There is no apparent trend for the residuals as a function of climate variables.

line 221: The authors did not *analyze* the dataset from Europe and North America in the same manner as the large dataset covering tropical continents.

The datasets are just plotted together. This only shows that datasets from northern

hemisphere are not outliers.

We agree with the Reviewer that this is what we have done. We do not believe a strict analysis in the manner that we have analysed the tropical/subtropical dataset is robust due to small sample sizes, so we have refrained from doing it. An analysis of the sort proposed by the Reviewer is also not sufficiently robust with inclusion of TRY data, as per our revised Supplemental Figure 8, because the data were collected and handled in very different ways from the main dataset (they are not species-at-site means, and they are not normalised to 25 °C, they are fit using potentially different algorithms for the FvCB model). Differences in the TRY dataset versus our data are detailed in the response to Reviewer #2 and in the revised figure caption for Supplemental Figure 8.

line 231-234: I get confused when looking at Supplementary Table 3. The intercept for Anet is not significant, while for Vcmax and Jmax P-value is bigger than 0.005...

The Reviewer might be confused because the analysis shown in Supplementary Table 3 is similar to, and somewhat redundant from, the analyses in Table 1/Supplementary Table 4 (now compressed into Table 1 as per Reviewer #2's comment). A key distinction is that the standardised major axis (SMA) analysis is free from assumptions regarding the distribution and precision of the independent variable, and the analyses in Supplementary Table 3 broadly support those in Table 1, except that this approach streamlines the analyses for testing for intercept differences and also shows the area-based analyses. Some modellers will be interested in the area-based results. The intercept differences in Supplementary Table 3 are relevant for the area-based measures V_{cmax} and J_{max} . We hope our explanation here has clarified the analyses for the Reviewer.

The last two lines of Supplementary Table 3 has both as "Mod. P", pls correct.

We have corrected this error and we apologise for the mistake.

Fig. 3 is nice. But how lowN/P and high N/P are defined? Please double check the caption. We apologise for presenting an unclear analysis of Fig. 3. In according with this comment and a similar one from Reviewer 1, we have reformulated Figure 3 to more directly address the issue based on clearly defined classes of leaf P_{mass} . The figure caption now more clearly identifies how P_{mass} classes are defined, as well as in the text, in accord with the Reviewer's question.

line 241-243: How do Fig. 1 and Fig. 2 show 'stronger effects of leaf P on Jmax'. I am confused.

The stronger effects that we refer to in this text pertain to Fig. 3, not Figs. 1 and 2. Hence the revised text is: "This supports the earlier evidence (Figs. 1 and 2) that there are strong effects of leaf P on Jmax, stronger than these effects are for Vcmax" (l. 197-198 in revision).

line 282-287: related to my previous comment, Fig. 2 is a less convincing way to show this than Supplementary Table 5. We have revised this to refer to Fig. 2 as well as Supplementary Tables 2 and 5, all of which collectively show similarities across taxa and continents.

References cited in our responses

- Achat, D. L. et al. (2016) Future challenges in coupled C-N-P cycle models for terrestrial ecosystems under global change: a review. *Biogeochemistry* 131, 173-202.
- Arrigo, K.R. (2005) Marine microorganisms and global nutrient cycles. *Nature* 437: 349-355
- Augusto, L. et al. (2017) Soil parent material-A major driver of plant nutrient limitations in terrestrial ecosystems. *Global Change Biology* 23, 3808-3824.
- Bracken, M.E.S., Hillebrand, H., Borer, E.T., Seabloom, E.W., Cebrian, J., Cleland, Elser, J.J., Gruner, D.S., Harpole, W.S., Ngai, J.T., Smith, J.E. (2015) Signatures of nutrient limitation and co-limitation: responses of autotroph internal nutrient concentrations to nitrogen and phosphorus additions. *Oikos* 124(2): 113-121
- Crous, KY, Ellsworth DS (2020) Probing the inner sanctum of leaf phosphorus: measuring the fractions of leaf P. *Plant and Soil* 454: 77-85.
- Goll, D. S. et al. (2012) Nutrient limitation reduces land carbon uptake in simulations with a model of combined carbon, nitrogen and phosphorus cycling. *Biogeosciences* 9, 3547-3569
- Hidaka, A. & Kitayama K. (2009) Divergent patterns of photosynthetic phosphorus-use efficiency versus nitrogen-use efficiency of tree leaves along nutrient-availability gradients *Journal of Ecology* 97, 984–991.
- Hu et al. (2022) Intercomparison of global terrestrial carbon fluxes estimated by MODIS and Earth system models. *Science of The Total Environment* 810(1), 152231
- Jonard et al (2015) Tree mineral nutrition is deteriorating in Europe. *Global Change Biology* 21: 418-430
- Lasslop, G. et al. (2010) Separation of net ecosystem exchange into assimilation and respiration using a light response curve approach: critical issues and global evaluation. *Global Change Biology* 16: 187-208, <https://doi.org/10.1111/j.1365-2486.2009.02041.x>
- Leuning, R. (2002) Temperature dependence of two parameters in a photosynthesis model. *Plant, Cell Environment* 25: 1205-1210.
- Pan, S., Tian, H., Dangal, S. R., Ouyang, Z., Lu, C., Yang, J., et al. (2015). Impacts of climate variability and extremes on global net primary production in the first decade of the 21st century. *Journal of Geographical Sciences*, 25(9), 1027–1044. <https://doi.org/10.1007/s11442-015-1217-4>
- Peñuelas et al (2013) Human-induced nitrogen–phosphorus imbalances alter natural and managed ecosystems across the globe. *Nature Commun* 4, 2934; <https://doi.org/10.1038/ncomms3934>
- Seiler, C., Melton, J. R., Arora, V. K., Sitch, S., Friedlingstein, P., Anthoni, P., et al. (2022). Are terrestrial biosphere models fit for simulating the global land carbon sink? *Journal of Advances in Modeling Earth Systems*, 14, e2021MS002946. <https://doi.org/10.1029/2021MS002946>

- Sprent, J. I. (2007). Evolving ideas of legume evolution and diversity: a taxonomic perspective on the occurrence of nodulation. *New Phytol.* 174, 11–25. doi: 10.1111/j.1469-8137.2007.02015.x
- Villareal, S. and Vargas, R. 2021. Representativeness of FLUXNET Sites Across Latin America. *J. Geophysical Research-Biogeosciences* 126(3), e2020JG006090, <https://doi.org/10.1029/2020JG006090>
- Wang, Y.P. and Goll, D. S. (2021) Modelling of land nutrient cycles: recent progress and future. *Faculty Reviews* 2021 10:(53)
- Wang, Z., Tian, H., Yang, J., Shi, H., Pan, S., Yao, Y., et al. (2020). Coupling of phosphorus processes with carbon and nitrogen cycles in the dynamic land ecosystem model: Model structure, parameterization, and evaluation in tropical forests. *Journal of Advances in Modeling Earth Systems*, 12, e2020MS002123. <https://doi.org/10.1029/2020MS002123>
- Werner, G. D. A., Cornwell, W. K., Sprent, J. I., Kattge, J., and Kiers, E. T. (2014). A single evolutionary innovation drives the deep evolution of symbiotic N₂-fixation in angiosperms. *Nat. Commun.* 5:4087. doi: 10.1038/ncomms5087
- Wieder, W. R., Cleveland, C. C., Smith, W. K., & Todd-Brown, K. (2015). Future productivity and carbon storage limited by terrestrial nutrient availability. *Nature Geoscience*, 8(6), 441–444. <https://doi.org/10.1038/ngeo2413>
- Wang, Y. P., Law, R., & Pak, B. (2010). A global model of carbon, nitrogen and phosphorus cycles for the terrestrial biosphere. *Biogeosciences*, 7(7), 2261–2282. <http://dx.doi.org/10.5194/bg-7-2261-2010>
- Yan, L. et al. (2019) Responses of foliar phosphorus fractions to soil age are diverse along a 2 Myr dune chronosequence. *New Phytologist* 223, 1621-1633.
- Yang, X. & Post, W. M. (2011) Phosphorus transformations as a function of pedogenesis: A synthesis of soil phosphorus data using Hedley fractionation method. *Biogeosciences* 8, 2907-2916.
- Yang, X., Ricciuto, D. M., Thornton, P. E., Shi, X., Xu, M., Hoffman, F., & Norby, R. J. (2019). The effects of phosphorus cycle dynamics on carbon sources and sinks in the Amazon region: a modeling study using ELM v1. *Journal of Geophysical Research: Biogeosciences*, 124, 3686–3698. <https://doi.org/10.1029/2019JG005082>

Response to reviewer comments, second round –

Reviewer #1 (Remarks to the Author):

The authors produced a revised version of their manuscript. Although all my initial comments were not taken into account, I am very satisfied by this new version, which was much improved. All in all, I have no major concern about this article and I recommend its publication. Congratulations for this nice study.

Suggestions:

The authors did an excellent work at clarifying the processes involved (lines 104-126 and Figure S1). Notably, the explanations about the V_{cmax} -N and J_{max} -P relationships will help much non-expert readers to understand the underlying biochemical functioning. However, one main result of this study was that, in fact, N and P interact to control both V_{cmax} and J_{max} (lines 185-194, 206-209, 241-249). This result is not fully in line with the fairly simple and straight forward initial description in the introduction. These interactive effects are well discussed (lines 362-381) but in a rather technical way. I suggest to slightly revise this section and improve the take-home message. Should the reader understand that the effect of P on V_{cmax} is an indirect consequence on its effect on J_{max} and subsequent regulation of the $V_{cmax}:J_{max}$ ratio? Please clarify.

By the way, because the studied relationships are based on an inter-specific comparison, one might argue that the observed effect of P on V_{cmax} is an artefact. Indeed, when comparing different plant species, it is well-established that the leaf N content and the leaf P content are highly correlated (e.g. Reich & Oleksyn, 2004, PNAS). Because of this relationship, the effect of P on V_{cmax} might be explained as a concomitant N-P variation and the role of the leaf N at controlling V_{cmax} . I am not fully aware of this particular literature but, if some existing articles present investigations of N-P-photosynthesis interactions within the same plant species, it would be good to briefly discuss the possible bias caused by N-P covariation among species.

Minor comments:

- line 111: remove one "in".
- Figure 3 and Table 1: four digits are enough for P values.

Reviewer #2 (Remarks to the Author):

The authors answered convincingly the questions I asked when reviewing the first version of the manuscript. Most of the proposed changes have been made.

As also pointed out by reviewer #1 I was surprised that more temperate and boreal sites could not be included in this study (and that P data from temperate and boreal regions are scarce in the TRY database) but this article should help initiate such studies.

The paper I referred to in my review regarding the response of forests to K availability was submitted to the same journal (NCOMMS) earlier this year but not yet published. However, I agree with the authors that mention of multi-element limitations is likely to distract from the main topic of the current paper.

I congratulate the authors for this study which should contribute to improve significantly the current global terrestrial models and I consider the manuscript now ready for publication.

Reviewer #3 (Remarks to the Author):

I found that the authors' response to reviewers' comments and revision of the manuscript satisfactory for me. The paper could be accepted for publication as it is.

“Convergence in phosphorus constraints to photosynthesis in forests around the world” by Ellsworth et al. – Responses to Reviewers #2

Responses to the Reviewers appear in blue text.

REVIEWERS' COMMENTS

Reviewer #1

The authors produced a revised version of their manuscript. Although all my initial comments were not taken into account, I am very satisfied by this new version, which was much improved. All in all, I have no major concern about this article and I recommend its publication. Congratulations for this nice study.

We are pleased that the Reviewer found the revisions acceptable, and we're grateful for his/her input and questions that improved the article content, including the suggestions below.

Suggestions:

The authors did an excellent work at clarifying the processes involved (lines 104-126 and Figure S1). Notably, the explanations about the V_{cmax-N} and J_{max-P} relationships will help much non-expert readers to understand the underlying biochemical functioning. However, one main result of this study was that, in fact, N and P interact to control both V_{cmax} and J_{max} (lines 185-194, 206-209, 241-249). This result is not fully in line with the fairly simple and straight forward initial description in the introduction. These interactive effects are well discussed (lines 362-381) but in a rather technical way. I suggest to slightly revise this section and improve the take-home message.

The Reviewer is correct: the simple model based on stoichiometric theory and past work in this area is not fully supported by our analysis, and this is an aspect where we push the field forward. The main goal of the manuscript was to uncover N and P interactions for photosynthesis and underlying photosynthetic biochemistry, and inform their use in terrestrial biosphere models. We state this on l. 454-464, as well as l. 311-315 and l. 328-333. This information should motivate future research into how leaf nutrients affect plant \$CO_2\$ assimilation, as well as refine and enhance terrestrial biosphere models.

Should the reader understand that the effect of P on V_{cmax} is an indirect consequence on its effect on J_{max} and subsequent regulation of the $V_{cmax}:J_{max}$ ratio? Please clarify.

We suggest this on l. 356-7 of the revised version by saying " functional balance between \$J_{max}\$ and \$V_{cmax}\$ is partially but not fully supported by Fig. 3". Beyond that, we are not comfortable providing a more definitive conclusion based on our data, understanding that the relationships we derived are correlative. We believe that more mechanistic work in this area would be needed to definitively say that the P effect on \$V_{cmax}\$ is a property of the effect on \$J_{max}\$ and its coupling to \$V_{cmax}\$. In fact, attempts to manipulate \$V_{cmax}\$ or \$J_{max}\$ separately using mutants and plants carrying gene knock-outs have not achieved larger differences in \$J_{max}:V_{cmax}\$ than we have shown with our data in Fig. 3 (see von Caemmerer et al. 1994, Rosenthal et al. 2011, Ruiz-Vera et al. 2022). So we regret that we cannot

answer the Reviewer's question with more certainty than we already have stated in the manuscript. We have presented evidence that the mode of action of P on V_{cmax} may operate through protein activation and phosphorylation (l. 358-361), a common mechanism of biochemical control, which is an alternative to the Reviewer's interpretation of the V_{cmax} -P correlation. We added text l. 381: "should consider ... the mechanistic implications of this in models."

By the way, because the studied relationships are based on an inter-specific comparison, one might argue that the observed effect of P on V_{cmax} is an artefact. Indeed, when comparing different plant species, it is well-established that the leaf N content and the leaf P content are highly correlated (e.g. Reich & Oleksyn, 2004, PNAS). Because of this relationship, the effect of P on V_{cmax} might be explained as a concomitant N-P variation and the role of the leaf N at controlling V_{cmax} .

Species composition can change markedly along geographic gradients in soil P. This is true within our global dataset or that of Reich & Oleksyn 2004, and at more local scales as documented by examples such as the Jurien Bay chronosequence in Western Australia (Laliberté et al. 2012, Laliberté et al. 2014), and soils of widely-contrasting nutrient concentrations within Amazonia (Coombes and Grubb 1996, Quesada et al. 2010). But why should observed trait correlations in cross-species analysis be necessarily "artefacts"? Conversely, why should only patterns seen within species be taken as definitive evidence of relationships between photosynthetic biochemistry and leaf P? We believe (but we're not sure) that this latter point is what the reviewer was suggesting, but it does not invalidate our analyses nor diminish the use of the P relationships in models, which themselves would want to incorporate the natural change in species composition across soils. Moreover, the very different species composition of different continents did not substantially change the leaf P relationships we have shown (see Fig. 2 and Supplementary Table 4).

We inserted l. 341 "bearing in mind that these relationships are across species" to provide the appropriate caveat based on the Reviewer's comment.

I am not fully aware of this particular literature but, if some existing articles present investigations of N-P-photosynthesis interactions within the same plant species, it would be good to briefly discuss the possible bias caused by N-P covariation among species.

See the previous response in relation to interpreting cross-species vs. within-species correlations. On another note, the issue of a "possible bias caused by N-P covariation" is referred to in statistical terms as 'multicollinearity'. Multicollinearity affects the coefficients and p-values, but it does not influence the predictions, precision of the predictions, and the goodness-of-fit statistics (Neter et al. 1996, Applied Linear Statistical Models, 4th Edition). As our primary goal is to inform predictions for models, we don't need to understand the role of each independent variable by reducing multicollinearity in our context. As a result, there may be bias in the coefficients and p-values of the relationships we present, but there is no problem with the predictions themselves that emerge from the statistical relationships we present. We have revised l. 246: "Moderate multicollinearity was observed with a significant correlation between N_{mass} and P_{mass} ($r^2 = 0.39$, $P < 0.0001$) across the dataset.

However, this does not affect predictability of V_{cmax} or J_{max} from N_{mass} and P_{mass} (see Neter et al. 1996)."

Minor comments:

- line 111: remove one "in". Done, thank you for catching this error.
- Figure 3 and Table 1: four digits are enough for P values. This has also been done.

Reviewer #2

The authors answered convincingly the questions I asked when reviewing the first version of the manuscript. Most of the proposed changes have been made.

We appreciate the input by the reviewer that improved the manuscript.

As also pointed out by reviewer #1 I was surprised that more temperate and boreal sites could not be included in this study (and that P data from temperate and boreal regions are scarce in the TRY database) but this article should help initiate such studies.

We do hope that our manuscript points out the lack of data and stimulates further research on plant N and P interactions on photosynthesis.

The paper I referred to in my review regarding the response of forests to K availability was submitted to the same journal (NCOMMS) earlier this year but not yet published. However, I agree with the authors that mention of multielement limitations is likely to distract from the main topic of the current paper.

It is regrettable that we're unable to cite this paper and we look forward to seeing it published soon.

I congratulate the authors for this study which should contribute to improve significantly the current global terrestrial models and I consider the manuscript now ready for publication.

We are grateful to the reviewer for insightful comments that caused us to look deeper into the analyses. Thank you.

Reviewer #3

I found that the authors' response to reviewers' comments and revision of the manuscript satisfactory for me. The paper could be accepted for publication as it is.

We thank the reviewer for his/her hard work and help improving the manuscript.

References

Coomes D, Grubb PJ. 1996. Amazonian caatinga and related communities at La Esmeralda, Venezuela: forest structure, physiognomy and floristics, and control by soil factors. *Vegetatio* 122: 167–191

Quesada CA, Lloyd J, Schwarz M, Patiño S, Baker TR et al. 2010. Chemical and physical properties of Amazon forest soils in relation to their genesis. *Biogeosciences*, 7, 1515–1541.

Laliberté E, Turner BL, Costes T, Pearse SJ, Wyrwoll K-H, Zemunik G, Lambers H. 2012. Experimental assessment of nutrient limitation along a 2-million year dune chronosequence in the south-western Australia biodiversity hotspot. *Journal of Ecology* 100: 631-642.

Laliberté E, Zemunik G, Turner BL. 2014. Environmental filtering explains variation in plant diversity along resource gradients. *Science* 345: 1602-1605.

Neter J, Kutner M, Wasserman W, Nachtsheim C 1996. *Applied Linear Statistical Models*, 4th Edition. McGraw-Hill, New York, USA. ISBN-13: 978-0256117363

Rosenthal DM, Lock AM, Khozaei M, Raines CA, Long SP. 2011. Over-expressing the C3 photosynthesis cycle enzyme sedoheptulose-1-7 bisphosphatase improves photosynthetic carbon gain and yield under fully open air CO₂ fumigation (FACE). *BMC Plant Biology* 2011, 11:123.

Ruiz-Vera UM, Acevedo-Siaca LG, Brown KL, Afamefule C, Gherli H, Simkin AJ, Long SP, Lawson T, Raines CA. 2022. Field-grown *ictB* tobacco transformants show no difference in photosynthetic efficiency for biomass relative to the wild type. *Journal of Experimental Botany*, erac193, <https://doi.org/10.1093/jxb/erac193>

von Caemmerer S, Evans JR, Hudson GS, Andrews TJ. 1994. The kinetics of ribulose-1,5-bisphosphate carboxylase/oxygenase in vivo inferred from measurements of photosynthesis in leaves of transgenic tobacco. *Planta* 195: 88–97.

“Convergence in phosphorus constraints to photosynthesis in forests around the world” by Ellsworth et al. – Responses to Reviewers #2

Responses to the Reviewers appear in blue text.

REVIEWERS' COMMENTS

Reviewer #1

The authors produced a revised version of their manuscript. Although all my initial comments were not taken into account, I am very satisfied by this new version, which was much improved. All in all, I have no major concern about this article and I recommend its publication. Congratulations for this nice study.

We are pleased that the Reviewer found the revisions acceptable, and we're grateful for his/her input and questions that improved the article content, including the suggestions below.

Suggestions:

The authors did an excellent work at clarifying the processes involved (lines 104-126 and Figure S1). Notably, the explanations about the V_{cmax-N} and J_{max-P} relationships will help much non-expert readers to understand the underlying biochemical functioning. However, one main result of this study was that, in fact, N and P interact to control both V_{cmax} and J_{max} (lines 185-194, 206-209, 241-249). This result is not fully in line with the fairly simple and straight forward initial description in the introduction. These interactive effects are well discussed (lines 362-381) but in a rather technical way. I suggest to slightly revise this section and improve the take-home message.

The Reviewer is correct: the simple model based on stoichiometric theory and past work in this area is not fully supported by our analysis, and this is an aspect where we push the field forward. The main goal of the manuscript was to uncover N and P interactions for photosynthesis and underlying photosynthetic biochemistry, and inform their use in terrestrial biosphere models. We state this on l. 454-464, as well as l. 311-315 and l. 328-333. This information should motivate future research into how leaf nutrients affect plant \$CO_2\$ assimilation, as well as refine and enhance terrestrial biosphere models.

Should the reader understand that the effect of P on V_{cmax} is an indirect consequence on its effect on J_{max} and subsequent regulation of the $V_{cmax}:J_{max}$ ratio? Please clarify.

We suggest this on l. 356-7 of the revised version by saying " functional balance between \$J_{max}\$ and \$V_{cmax}\$ is partially but not fully supported by Fig. 3". Beyond that, we are not comfortable providing a more definitive conclusion based on our data, understanding that the relationships we derived are correlative. We believe that more mechanistic work in this area would be needed to definitively say that the P effect on \$V_{cmax}\$ is a property of the effect on \$J_{max}\$ and its coupling to \$V_{cmax}\$. In fact, attempts to manipulate \$V_{cmax}\$ or \$J_{max}\$ separately using mutants and plants carrying gene knock-outs have not achieved larger differences in \$J_{max}:V_{cmax}\$ than we have shown with our data in Fig. 3 (see von Caemmerer et al. 1994, Rosenthal et al. 2011, Ruiz-Vera et al. 2022). So we regret that we cannot

answer the Reviewer's question with more certainty than we already have stated in the manuscript. We have presented evidence that the mode of action of P on V_{cmax} may operate through protein activation and phosphorylation (l. 358-361), a common mechanism of biochemical control, which is an alternative to the Reviewer's interpretation of the V_{cmax} -P correlation. We added text l. 381: "should consider ... the mechanistic implications of this in models."

By the way, because the studied relationships are based on an inter-specific comparison, one might argue that the observed effect of P on V_{cmax} is an artefact. Indeed, when comparing different plant species, it is well-established that the leaf N content and the leaf P content are highly correlated (e.g. Reich & Oleksyn, 2004, PNAS). Because of this relationship, the effect of P on V_{cmax} might be explained as a concomitant N-P variation and the role of the leaf N at controlling V_{cmax} .

Species composition can change markedly along geographic gradients in soil P. This is true within our global dataset or that of Reich & Oleksyn 2004, and at more local scales as documented by examples such as the Jurien Bay chronosequence in Western Australia (Laliberté et al. 2012, Laliberté et al. 2014), and soils of widely-contrasting nutrient concentrations within Amazonia (Coombes and Grubb 1996, Quesada et al. 2010). But why should observed trait correlations in cross-species analysis be necessarily "artefacts"? Conversely, why should only patterns seen within species be taken as definitive evidence of relationships between photosynthetic biochemistry and leaf P? We believe (but we're not sure) that this latter point is what the reviewer was suggesting, but it does not invalidate our analyses nor diminish the use of the P relationships in models, which themselves would want to incorporate the natural change in species composition across soils. Moreover, the very different species composition of different continents did not substantially change the leaf P relationships we have shown (see Fig. 2 and Supplementary Table 4).

We inserted l. 341 "bearing in mind that these relationships are across species" to provide the appropriate caveat based on the Reviewer's comment.

I am not fully aware of this particular literature but, if some existing articles present investigations of N-P-photosynthesis interactions within the same plant species, it would be good to briefly discuss the possible bias caused by N-P covariation among species.

See the previous response in relation to interpreting cross-species vs. within-species correlations. On another note, the issue of a "possible bias caused by N-P covariation" is referred to in statistical terms as 'multicollinearity'. Multicollinearity affects the coefficients and p-values, but it does not influence the predictions, precision of the predictions, and the goodness-of-fit statistics (Neter et al. 1996, Applied Linear Statistical Models, 4th Edition). As our primary goal is to inform predictions for models, we don't need to understand the role of each independent variable by reducing multicollinearity in our context. As a result, there may be bias in the coefficients and p-values of the relationships we present, but there is no problem with the predictions themselves that emerge from the statistical relationships we present. We have revised l. 246: "Moderate multicollinearity was observed with a significant correlation between N_{mass} and P_{mass} ($r^2 = 0.39$, $P < 0.0001$) across the dataset.

However, this does not affect predictability of V_{\max} or J_{\max} from N_{mass} and P_{mass} (see Neter et al. 1996)."

Minor comments:

- line 111: remove one "in". Done, thank you for catching this error.
- Figure 3 and Table 1: four digits are enough for P values. This has also been done.

Reviewer #2

The authors answered convincingly the questions I asked when reviewing the first version of the manuscript. Most of the proposed changes have been made.

We appreciate the input by the reviewer that improved the manuscript.

As also pointed out by reviewer #1 I was surprised that more temperate and boreal sites could not be included in this study (and that P data from temperate and boreal regions are scarce in the TRY database) but this article should help initiate such studies.

We do hope that our manuscript points out the lack of data and stimulates further research on plant N and P interactions on photosynthesis.

The paper I referred to in my review regarding the response of forests to K availability was submitted to the same journal (NCOMMS) earlier this year but not yet published. However, I agree with the authors that mention of multielement limitations is likely to distract from the main topic of the current paper.

It is regrettable that we're unable to cite this paper and we look forward to seeing it published soon.

I congratulate the authors for this study which should contribute to improve significantly the current global terrestrial models and I consider the manuscript now ready for publication.

We are grateful to the reviewer for insightful comments that caused us to look deeper into the analyses. Thank you.

Reviewer #3

I found that the authors' response to reviewers' comments and revision of the manuscript satisfactory for me. The paper could be accepted for publication as it is.

We thank the reviewer for his/her hard work and help improving the manuscript.

References

Coomes D, Grubb PJ. 1996. Amazonian caatinga and related communities at La Esmeralda, Venezuela: forest structure, physiognomy and floristics, and control by soil factors. *Vegetatio* 122: 167–191

Quesada CA, Lloyd J, Schwarz M, Patiño S, Baker TR et al. 2010. Chemical and physical properties of Amazon forest soils in relation to their genesis. *Biogeosciences*, 7, 1515–1541.

Laliberté E, Turner BL, Costes T, Pearse SJ, Wyrwoll K-H, Zemunik G, Lambers H. 2012. Experimental assessment of nutrient limitation along a 2-million year dune chronosequence in the south-western Australia biodiversity hotspot. *Journal of Ecology* 100: 631-642.

Laliberté E, Zemunik G, Turner BL. 2014. Environmental filtering explains variation in plant diversity along resource gradients. *Science* 345: 1602-1605.

Neter J, Kutner M, Wasserman W, Nachtsheim C 1996. *Applied Linear Statistical Models*, 4th Edition. McGraw-Hill, New York, USA. ISBN-13: 978-0256117363

Rosenthal DM, Lock AM, Khozaei M, Raines CA, Long SP. 2011. Over-expressing the C3 photosynthesis cycle enzyme sedoheptulose-1-7 bisphosphatase improves photosynthetic carbon gain and yield under fully open air CO₂ fumigation (FACE). *BMC Plant Biology* 2011, 11:123.

Ruiz-Vera UM, Acevedo-Siaca LG, Brown KL, Afamefule C, Gherli H, Simkin AJ, Long SP, Lawson T, Raines CA. 2022. Field-grown *ictB* tobacco transformants show no difference in photosynthetic efficiency for biomass relative to the wild type. *Journal of Experimental Botany*, erac193, <https://doi.org/10.1093/jxb/erac193>

von Caemmerer S, Evans JR, Hudson GS, Andrews TJ. 1994. The kinetics of ribulose-1,5-bisphosphate carboxylase/oxygenase in vivo inferred from measurements of photosynthesis in leaves of transgenic tobacco. *Planta* 195: 88–97.